JCB Journal of Cell Biology

# The deubiquitinase USP45 inhibits autophagy through actin regulation by Coronin 1B

Yuchieh Jay Lin[1,2,3] [ID], Li-Ting Huang[1,2] [ID], Po-Yuan Ke[4] [ID], and Guang-Chao Chen[1,2,3] [ID]

The autophagy–lysosomal system comprises a highly dynamic and interconnected vesicular network that plays a central role in maintaining proteostasis and cellular homeostasis. In this study, we uncovered the deubiquitinating enzyme (DUB), dUsp45/USP45, as a key player in regulating autophagy and lysosomal activity in *Drosophila* and mammalian cells. Loss of *dUsp45/USP45* results in autophagy activation and increased levels of V-ATPase to lysosomes, thus enhancing lysosomal acidification and function. Furthermore, we identified the actin-binding protein Coronin 1B (Coro1B) as a substrate of USP45. USP45 interacts with and deubiquitinates Coro1B, thereby stabilizing Coro1B levels. Notably, the ablation of USP45 or Coro1B promotes the formation of F-actin patches and the translocation of V-ATPase to lysosomes in an N-WASP-dependent manner. Additionally, we observed positive effects of dUsp45 depletion on extending lifespan and ameliorating polyglutamine (polyQ)-induced toxicity in *Drosophila*. Our findings highlight the important role of dUsp45/USP45 in regulating lysosomal function by modulating actin structures through Coro1B.

## Introduction

Macroautophagy (hereafter autophagy) is a highly conserved catabolic pathway that plays a pivotal role in maintaining cellular homeostasis. The autophagic process is comprised of multiple steps from the formation of double-membrane autophagosomes, which engulf cytosolic macromolecules such as protein aggregates and damaged organelles to fuse with the lysosomes (Hu and Reggiori, 2022; Yamamoto et al., 2023). Lysosomes are membrane-bound acidic organelles that contain an array of luminal hydrolases essential for the degradation and recycling of cell materials (Ballabio and Bonifacino, 2020). The acidic environment within lysosomes is mainly generated by the vacuolar ATPase (V-ATPase) that pumps protons from the cytoplasm to the lumen of lysosomes (Collins and Forgac, 2020). The acidic pH–activated lysosomal hydrolases break down enclosed cargos into building-block molecules, including amino acids, monosaccharides, and free fatty acids, which could subsequently be exported from lysosomes for metabolic utilization.

Besides its role as the terminal digestive center of the autophagic pathway, the lysosome has emerged as a key player in cellular signaling and nutrient sensing. Under nutrient-rich conditions, the nutrient sensor mechanistic target of rapamycin (mTOR) complex 1 (mTORC1) is recruited to lysosomes by the heterodimeric Rag GTPases, which are tethered to the lysosomal surface via the heteropentameric Ragulator protein complex (Sancak et al., 2010). Moreover, in a V-ATPase-dependent manner, the Ragulator complex collaborates with the arginine transporter SCL38A9 to function as a guanine nucleotide exchange factor (GEF) for Rag GTPases, facilitating mTORC1 lysosomal localization and activation (Rebsamen et al., 2015). The lysosomal-localized mTORC1 is further activated by the Rheb GTPase (Angarola and Ferguson, 2019; Hao et al., 2018). In addition to mTORC1, the active Rag GTPases interact and recruit the MiT/TFE family transcription factor EB (TFEB) to lysosomes (Martina and Puertollano, 2013). TFEB is a master regulator for lysosome biogenesis by inducing lysosomal and autophagy-related gene expression (Settembre et al., 2011). However, in nutrient-rich environments, mTORC1-mediated phosphorylation of TFEB impairs its nuclear translocation, which reduces autophagy and lysosome biogenesis (Puertollano et al., 2018; Settembre et al., 2012).

The assembly and disassembly of actin filaments by actin regulators play crucial roles in a wide range of cellular activities, including autophagy. The actin capping protein CapZ (CAPZB) and the WASP homolog associated with actin Golgi membranes and microtubules (WHAMM)–Arp2/3 complex–mediated branched actin networks are involved in shaping the phagophore membrane and facilitating the expansion and closure of the autophagosomal membrane during the biogenesis of autophagosomes (Kast et al., 2015; Mi et al., 2015). Moreover, several actin-based motors, including myosins I, II, and VI, have been

[1]Institute of Biological Chemistry, Academia Sinica, Taipei, Taiwan;   [2]Institute of Biochemical Sciences, College of Life Science, National Taiwan University, Taipei, Taiwan;   [3]Chemical Biology and Molecular Biophysics, Taiwan International Graduate Program, Academia Sinica, Taipei, Taiwan;   [4]Department of Biochemistry & Molecular Biology and Graduate Institute of Biomedical Sciences, College of Medicine, Chang Gung University, Taoyuan, Taiwan.

Correspondence to Guang-Chao Chen: gcchen@gate.sinica.edu.tw.



involved in guiding the movement of cargos along actin filaments for phagophore expansion, autophagosome maturation, and fusion with lysosomes (Brandstaetter et al., 2014; Tang et al., 2011; Tumbarello et al., 2012). In selective autophagy, cargo recognition and selective autophagy receptors may interact with actin filaments, facilitating the recruitment of autophagosomal membranes to the cargo for degradation (Tumbarello et al., 2013, 2015). Moreover, several signaling pathways are found to regulate actin organization and autophagy. For example, the Rho family small GTPases are master regulators of actin dynamics. Recent research has demonstrated that RhoA overexpression or constitutively active RhoA suppresses mTORC1 signaling, which leads to the activation of autophagic processes (Gordon et al., 2014). In contrast, the inhibition of Rac1 or deletion of Rac1 attenuates mTORC1 activation, thereby enhancing the autophagic pathway (Saci et al., 2011). However, the precise mechanism by which actin regulates V-ATPase activity and autolysosomal function remains elusive.

Ubiquitination is one of the most notable posttranslational modifications (PTMs) involved in the regulation of the autophagic process, ranging from cargo recognition to the nucleation of autophagosomes and fusion with the lysosome (Chen et al., 2019). The ubiquitination process is reversibly regulated by E3 ligases and deubiquitinases (DUBs) by the addition or removal of ubiquitin on substrates, respectively. While many autophagy regulators have been identified as substrates for ubiquitin E3 ligases, the involvement of DUBs in autophagy remains less well-defined (Csizmadia and Lőw, 2020). In this study, we reported that dUsp45/USP45 acts as a negative regulator of autophagy and lysosomal function in both *Drosophila* and mammalian cells. dUsp45/USP45 impairs lysosomal acidification and disrupts the localization of V-ATPases to lysosomes by modulating actin structures. USP45 interacted with and deubiquitinated the actin-regulatory protein Coronin 1B (Coro1B) to promote actin turnover. Ablation of USP45 led to an increase in the formation of F-actin patches associated with lysosomes and elevated levels of lysosomal V-ATPases, resulting in heightened autolysosomal activity. Our study further showed that loss of *dUsp45* extended the lifespan of adult flies and ameliorated polyglutamine (polyQ)-induced eye degeneration. Collectively, our observations reveal the critical function of USP45 in the regulation of actin structures and lysosomal activity.

## Results

### *Drosophila* Usp45 acts as a negative regulator of autophagy

We and others have recently conducted an RNA interference (RNAi) screen for DUBs involved in the regulation of autophagy in *Drosophila* larval fat body (Jacomin et al., 2015; Pai et al., 2023). Interestingly, we found that the knockdown of *Drosophila* Usp45 (dUsp45) using two independent RNAi lines caused a dramatic increase of small Atg8a puncta under well-fed conditions (Fig. 1, A and C). Moreover, coexpression of dUsp45 markedly reduced the number of Atg8a puncta in dUsp45-depleted larval fat body cells (Fig. S1, A and B), suggesting that Atg8a puncta accumulation is not due to an off-target effect. dUsp45 belongs to the ubiquitin-specific protease family and its

mammalian homolog, USP45, has been shown to regulate DNA damage and cell migration (Conte et al., 2018; Perez-Oliva et al., 2015); however, the role of USP45 in autophagy remains unknown. Because the accumulation of mCherry-Atg8a puncta in *dUsp45*-knockdown larval fat body cells could indicate either autophagic activation or impaired autophagy flux, we checked the effects of dUsp45 depletion on autophagic flux by feeding the larvae with chloroquine (CQ), which prevents the fusion of autophagosomes and lysosomes. As shown in Fig 1, B and C, CQ treatment dramatically increased the size and number of mCherry-Atg8a puncta in dUsp45 depleted cells, compared with controls. The effects of *dUsp45* knockdown on autophagic flux were further examined using the tandem fluorescent-tagged GFP-mCherry-Atg8a reporter. The tandem reporter displays both green and red fluorescence signals when localized to autophagosomes; however, the GFP fluorescence is quickly quenched in acidic autolysosomes. We found that the knockdown of *dUsp45* caused a dramatic increase in the number of the total Atg8a puncta and the ratio of autolysosomes (red, mCherry$^+$ GFP$^-$) to total Atg8a puncta compared with controls (Fig. 1, D and E). These results suggest that depletion of dUsp45 promotes autophagosome formation and autophagic flux.

Previous studies have shown that the ecdysone-PI3K pathway induces autophagy activation during *Drosophila* late larval development and metamorphosis (Rusten et al., 2004). The transcriptional profile of *dUsp45* across different stages of *Drosophila* development shows high expression during embryogenesis and moderate to low expression during the larval stages (Graveley et al., 2011). Consistently, we observed a reduction in *dUsp45* gene expression in the fat bodies of wandering third-instar (wL3) larvae compared with second-instar (L2) larvae (Fig. 1 F). We next examined whether *dUsp45* plays a role in the regulation of developmental autophagy. As observed in L2 larvae, *dUsp45* depletion caused smaller autolysosomes in the fat body cells of wL3 larvae compared with control cells (Fig. S1, C and D). To further confirm dUsp45 function, wild-type (WT) dUsp45 and the catalytically inactive mutant dUsp45-C315A (CA) were expressed in the larval fat bodies (Fig. S1 E). Immunofluorescence analysis revealed that dUsp45-WT, but not the catalytic mutant dUsp45-CA, caused a decrease in the number of Atg8a puncta at the wL3 larval stage (Fig. 1, G and H). Notably, the size of the Atg8a puncta was significantly enlarged in cells expressing dUsp45-WT compared with controls (Fig. 1, G and H). Transmission electron microscopy (TEM) analysis revealed that expression of dUsp45-WT resulted in the formation of enlarged autolysosomes (Fig. S1, F and G). Moreover, immunofluorescence and immunoblotting analyses showed an accumulation of the autophagy substrate p62/Ref2P in dUsp45-WT, but not in control or dUsp45-CA expressing cells (Fig. 1, I and J; and Fig. S1, H and I), suggesting that dUsp45-WT impairs autophagy. Previous studies have demonstrated the critical role of autophagy in *Drosophila* metamorphosis (Tracy and Baehrecke, 2013). Notably, overexpression of dUsp45-WT resulted in a dramatic delay in pupariation, whereas dUsp45 depletion accelerated pupariation in wandering larvae (Fig. S1 J). Together, these results indicate that dUsp45 plays a negative role in regulating developmental autophagy in *Drosophila* wandering wL3 larval fat body tissues.

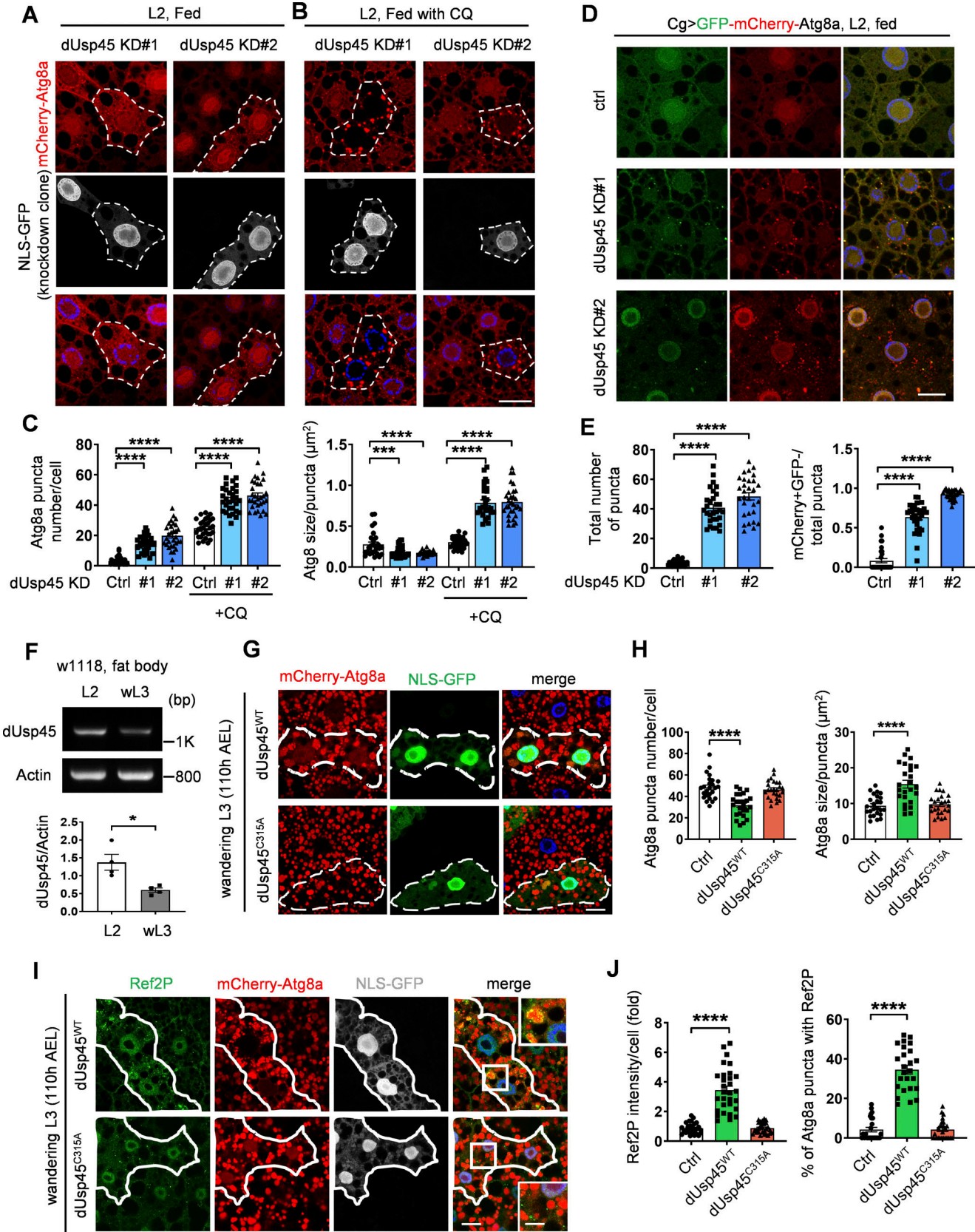

Figure 1. **dUsp45 depletion enhances autophagosome formation and autophagic flux. (A and B)** The clonal knockdown of *dUsp45* (GFP positive) in the larval fat bodies using the flp-out system resulted in an elevated presence of mCherry-Atg8a puncta, compared with controls (GFP negative). Secondary instar larvae (L2) were incubated in normal food (A) or with chloroquine (CQ, 10 mg/ml) for 6 h (B). Knockdown clones are indicated by the dashed line. Scale bar: 20

μm. **(C)** Quantification results of the number and size of Atg8a puncta were depicted for conditions (A and B). Data are shown as mean ± SEM, $n$ = 3, ≥ 30 cells. **(D)** Confocal microscopy analysis of autophagic flux, as determined by the expression of GFP-mCherry-Atg8a with the fat body-specific *Cg-Gal4* driver, in control (Ctrl) and *dUsp45* knockdown larvae. Scale bar, 20 μm. **(E)** Quantification of total Atg8a puncta and ratio of autolysosomes (GFP⁻ mCherry⁺) to total Atg8a puncta per cell in D; data are shown as mean ± SEM, $n$ = 3, ≥ 30 cells. **(F)** RT-PCR analysis of *dUsp45* mRNA levels in the fat body of second instar (L2) larvae and wandering third-instar (wL3) larvae. Quantification results indicated *dUsp45* expression levels normalized to *Actin*. **(G)** The clonal expression of dUsp45-WT (GFP positive), but not catalytic mutant dUsp45-C315A (GFP positive), in the wL3 larval fat bodies using the flp-out system resulted in an enlarged size of Atg8a puncta, compared with controls (GFP negative). The clones were indicated by the dashed line. Scale bar: 20 μm. **(H)** Quantification of the number and size of Atg8a puncta per cell in G; data shown as mean ± SEM, $n$ = 3, ≥ 25 cells. **(I)** The clonal expression of dUsp45-WT, but not catalytic mutant dUsp45-C315A, in the wL3 larval fat bodies resulted in increased Ref2P signal, compared with controls. The clones were indicated by white lines. Scale bar: 20 μm (original) and 10 μm (zoom-in). **(J)** Quantification of Ref2P intensity per cell and percentage of the Atg8a puncta colocalized with Ref2P dots. Data are shown as mean ± SEM, $n$ = 3, ≥ 25 cells. Significance was determined using one-way ANOVA and Dunnett's multiple comparisons test (C, E, H, and J), and Student's $t$ test (F); *$P < 0.05$; **$P < 0.01$; ****$P < 0.0001$. Source data are available for this figure: SourceData F1.

## dUsp45 inhibits lysosome acidification and lysosomal function by disrupting V-ATPase translocation to lysosomes

Since dUsp45 negatively regulates autophagy and impairs autophagic degradation, we first checked whether dUsp45 may inhibit autophagosome–lysosome fusion. Confocal microscopy of wL3 larval fat bodies revealed a high degree of colocalization between autophagosomes (mCherry-Atg8a) and lysosomes (GFP-LAMP1) in control and dUsp45-WT expressing larval fat body cells (Fig. 2, A and B), suggesting that dUsp45 did not affect the autophagosome–lysosome fusion. Next, we examined whether dUsp45 regulates lysosomal function by staining the larval fat body with LysoTracker and Magic Red, which stain for cellular acidic lysosomes and cathepsin activity, respectively. Immunofluorescence analysis revealed that expression of dUsp45 dramatically reduced the number of acidic lysosomes, whereas depletion of dUsp45 resulted in increased LysoTracker puncta compared with controls (Fig. 2, C and D). Similarly, dUsp45 knockdown caused significantly increased levels of Magic Red fluorescence (Fig. 2 E) and cathepsin-L expression (Fig. 2, F and G). These results together indicate that dUsp45 plays a critical role in regulating lysosomal function by attenuating lysosome acidification and cathepsin activity.

The vacuolar H⁺-ATPase (V-ATPase) is an ATP-dependent proton pump that is comprised of the multisubunit peripheral V1 domain and the integral V0 domain (Vasanthakumar and Rubinstein, 2020). The translocation and assembly of V1 and V0 domains on lysosomes are critical for V-ATPase function to transport protons from cytosol to lysosomal lumen and maintain lysosomal acidification (Collins and Forgac, 2018; Song et al., 2020). We thus checked whether dUsp45 may regulate the localization of V-ATPase to lysosomes. Immunofluorescence analysis revealed that VhaSFD-GFP and Vha13-GFP, which are V1 subunits of V-ATPase, were highly enriched surrounding Atg8a-positive autolysosomes in wL3 larval fat body cells (Fig. 2, H and I; and Fig. S1, K and L). Strikingly, the localization of VhaSFD-GFP and Vha13-GFP on autolysosomes was dramatically decreased in dUsp45-WT expressing larval fat body cells (Fig. 2, H and I; and Fig. S1, K and L), suggesting that dUsp45 inhibits lysosomal acidification by interfering with the translocation of the V-ATPase complex to autolysosomes.

## USP45 negatively regulates autophagy and lysosomal activity

The mammalian USP45 is highly homologous to the *Drosophila* dUsp45 (Fig. 3 A). We therefore investigated whether mammalian USP45 also played a conserved role in the regulation of autophagy. First, we checked the localization of USP45 in mammalian cells. Immunofluorescent images showed that USP45 was highly colocalized with several endosomal markers, including early endosome (EEA1), late endosome (Rab7), and lysosome (LAMP1), but relatively low with autophagosome (LC3) under normal conditions (Fig. S2, A and B). Next, the stable *USP45*-knockdown HeLa cells were generated to check their function in autophagy (Fig. S2 C). Ablation of USP45 led to a significant increase in LC3 puncta and LC3-II/GAPDH levels, both in the presence and absence of the lysosomal inhibitor bafilomycin A1 (BafA1), compared with controls (Fig. 3, B–E). Furthermore, the levels of the autophagy substrate p62/SQSTM1 were markedly reduced in USP45-depleted cells (Fig. 3, D and E). A tandem fluorescent-tagged GFP-RFP-LC3 reporter assay further revealed that *USP45* knockdown caused a dramatic increase in the total number of LC3 puncta and the ratio of autolysosomes (RFP⁺ GFP⁻) to total LC3 puncta compared with controls (Fig. 3, F and G). These results suggest that, similar to its *Drosophila* counterpart, mammalian USP45 also negatively regulates autophagosome formation and autophagy flux.

To investigate the mechanism by which USP45 activates autophagy, we examined the effects of USP45 depletion on the expression of key autophagy-regulating proteins. While mTORC1 protein expression remained unaffected, the phosphorylation of its downstream target, p70 ribosomal protein S6 kinase (S6K), was reduced in USP45-depleted cells, indicating a decrease in mTOR activity (Fig. 3, H and I). This reduction in mTOR activity is known to be associated with autophagy activation. Additionally, USP45 depletion led to increased expression of ULK1, VPS34, and Beclin1 (Fig. 3, J and K), which are key regulators of autophagy initiation and autophagosome formation. mTORC1 phosphorylates and prevents the nuclear localization of the TFEB, a member of the MiT-TFE family that drives the expression of autophagy and lysosomal genes (Napolitano and Ballabio, 2016). Notably, we observed a significant increase in GFP-TFEB nuclear localization in USP45-depleted cells (Fig. 3, L and M), suggesting that USP45 regulates autophagy activation by modulating mTOR activity.

We next checked whether mammalian USP45 also affected lysosomal activity. The lysosomal protease cathepsin enzyme activity is regulated by pH. Inside lysosomes, the procathepsins are cleaved to form mature active cathepsins upon acidification

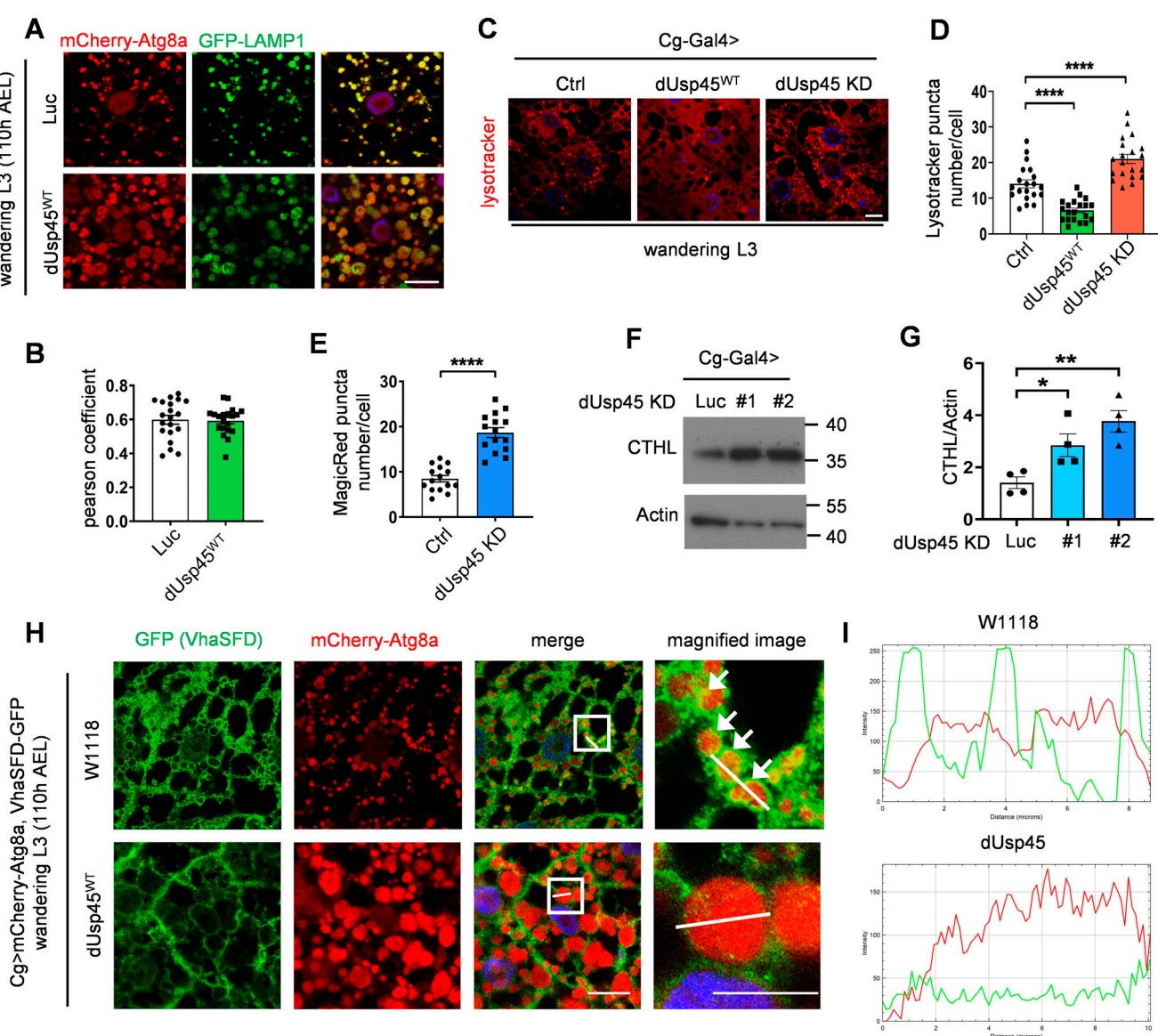

**Figure 2. dUsp45 overexpression impairs lysosomal acidification and V-ATPase lysosomal localization. (A)** Confocal microscopy analysis showed colocalization of mCherry-Atg8a and GFP-LAMP1 in control or dUsp45^WT expressing wL3 larval fat body cells. Scale bar, 20 μm. **(B)** Quantification of the colocalization of Atg8a and LAMP1 in A. Pearson's correlation coefficient was analyzed by ImageJ. Data are shown as mean ± SEM, n = 3, ≥ 20 cells. **(C)** Confocal microscopy analysis of wL3 larval fat body cells expressing *Luc* (Ctrl), *dUsp45^WT*, or *dUsp45^RNAi* with *Cg-Gal4* driver and stained with the fluorescent dye LysoTracker Red. Scale bar, 20 μm. **(D)** Quantification of the number of LysoTracker-positive dots per cell in C; data shown as mean ± SEM, n = 3, ≥ 20 cells. **(E)** Quantification of the number of MagicRed puncta per cell in *dUsp45* knockdown wL3 larval fat body cells. Data are shown as mean ± SEM, n = 3, ≥ 15 cells. **(F)** Western blot analysis of Cathepsin-L (CTHL) expression levels in the larval fat bodies expressing *Luc* (Ctrl) or *dUsp45^RNAi* under the control of *Cg-Gal4*. **(G)** Quantification of Cathepsin-L expression normalized to Actin in F. Data shown as mean ± SEM of four independent experiments. **(H)** Confocal microscopy analysis of localization of VhaSFD (GFP positive) and mCherry-Atg8a in control and dUsp45^WT overexpression wL3 larval fat body cells. The arrows showed the colocalized signals of VhaSFD and Atg8a. The scale bars showed 20 μm (original) and 10 μm (zoom-in). **(I)** Line-scan profiles of fluorescence intensity for mCherry-Atg8a and GFP-VhaSFD along the white line in H. Significance was determined using one-way ANOVA and Dunnett's multiple comparisons test (D and G), and Student's *t* test (B and E); *P < 0.05; **P < 0.01; ****P < 0.0001. Source data are available for this figure: SourceData F2.

(Stoka et al., 2016). Immunoblotting analysis revealed elevated levels of mature cathepsin-L in *USP45* knockdown cells (Fig. 4, A and B). In addition, the DQ-BSA assay was performed to assess the lysosomal degradation function. As shown in Fig. 4, C and D, the number of DQ-BSA puncta was dramatically increased in USP45-depleted cells, suggesting an antagonizing role of USP45 in the regulation of lysosomal function. Consistent with our

findings in *Drosophila*, elevated levels of LysoTracker staining were observed in USP45-depleted cells compared with controls, which suggests an increase in lysosomal acidity (Fig. S2, D and E). We then investigated whether USP45 could modulate the localization of V-ATPase on endolysosomes. Interestingly, the ablation of USP45 significantly enhanced the localization of the V-ATPase V0 subunit d (ATP6V0D) at late endosomes (Rab 7)

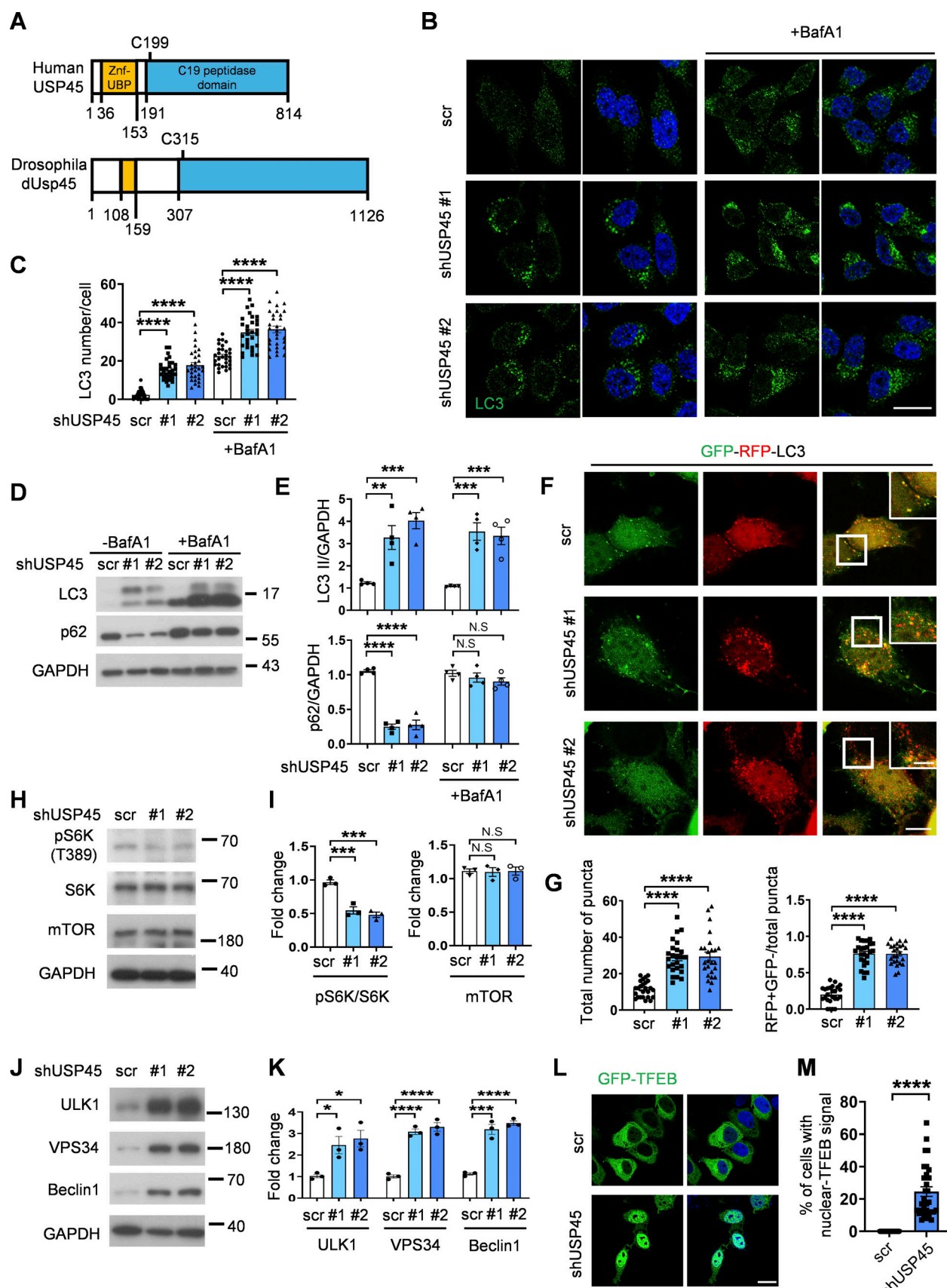

Figure 3. **Mammalian USP45 negatively regulates autophagy. (A)** Schematic presentation of the domain structures of human USP45 and *Drosophila* dUsp45. **(B)** Immunofluorescence analysis of LC3 puncta in control (scr) or *USP45* knockdown HeLa cells with or without bafilomycin A1 (BafA1) treatment.

Scale bar, 20 µm. **(C)** Quantification of the number of LC3 puncta in control and *USP45* knockdown cells in B; data shown as mean ± SEM, n = 3, ≥ 30 cells. **(D)** Western blot analysis of LC3 and p62 levels in control and USP45 depleted HeLa cells with or without BafA1 treatment. **(E)** Quantification of LC3 and p62 expression normalized to GAPDH in D. Data shown as mean ± SEM of four independent experiments. **(F)** Confocal microscopy analysis of autophagy flux in mRFP-EGFP-LC3 transfected control or *USP45* knockdown HeLa cells. Scale bar showed 10 µm (original) and 5 µm (zoom-in). **(G)** Quantification of the number of total LC3 puncta and the ratio of autolysosomes (RFP$^+$GFP$^-$) to total LC3 puncta. Data as shown by mean ± SEM, n = 3, ≥25 cells/condition. **(H)** Western blot analysis of phospho-p70 S6 kinase (pS6K), total p70 S6 kinase (S6K), and mTOR expression levels in control and *USP45* knockdown HeLa cells. **(I)** Quantification of phospho-S6K normalized to total S6K, and mTOR levels normalized to GAPDH in H. Data shown as mean ± SEM of three independent experiments. **(J)** Western blot analysis of ULK1, VPS34, and Beclin1 expression levels in control and *USP45* knockdown HeLa cells. **(K)** Quantification of ULK1, VPS34, and Beclin1 levels normalized to GAPDH in J. Data shown as mean ± SEM of three independent experiments. **(L)** Immunofluorescent analysis of GFP-TFEB localization in control and *USP45* knockdown HeLa cells. Scale bar, 20 µm. **(M)** Quantification of the percentage of cells with nuclear GFP-TFEB signal. Data are shown by mean ± SEM, n = 30 images, ≥5 cells/image. Significance was determined using one-way ANOVA and Dunnett's multiple comparisons test (C, E, G, I, and K), and Student's t test (M); *P < 0.05; **P < 0.01; ***P < 0.001; ****P < 0.0001; NS not significant. Source data are available for this figure: SourceData F3.

and lysosomes (LAMP2), while having no effects on its localization at early endosomes (EEA1) (Fig. 4, E and F). Similarly, the cytosolic V-ATPase V1 complex (ATP6V1H) exhibited notably higher colocalization with LAMP1-positive lysosomes in USP45-depleted cells compared with controls (Fig. S2, F and G). Furthermore, through lysosomal immunoprecipitation (LysoIP) assays, we observed increased levels of V1B and V0D in the lysosomal fraction of USP45 knockdown cells (Fig. S2, H and I). These findings collectively suggest that mammalian USP45 plays an evolutionarily conserved role in suppressing lysosomal acidification by interfering with the translocation of V-ATPase to endolysosomes.

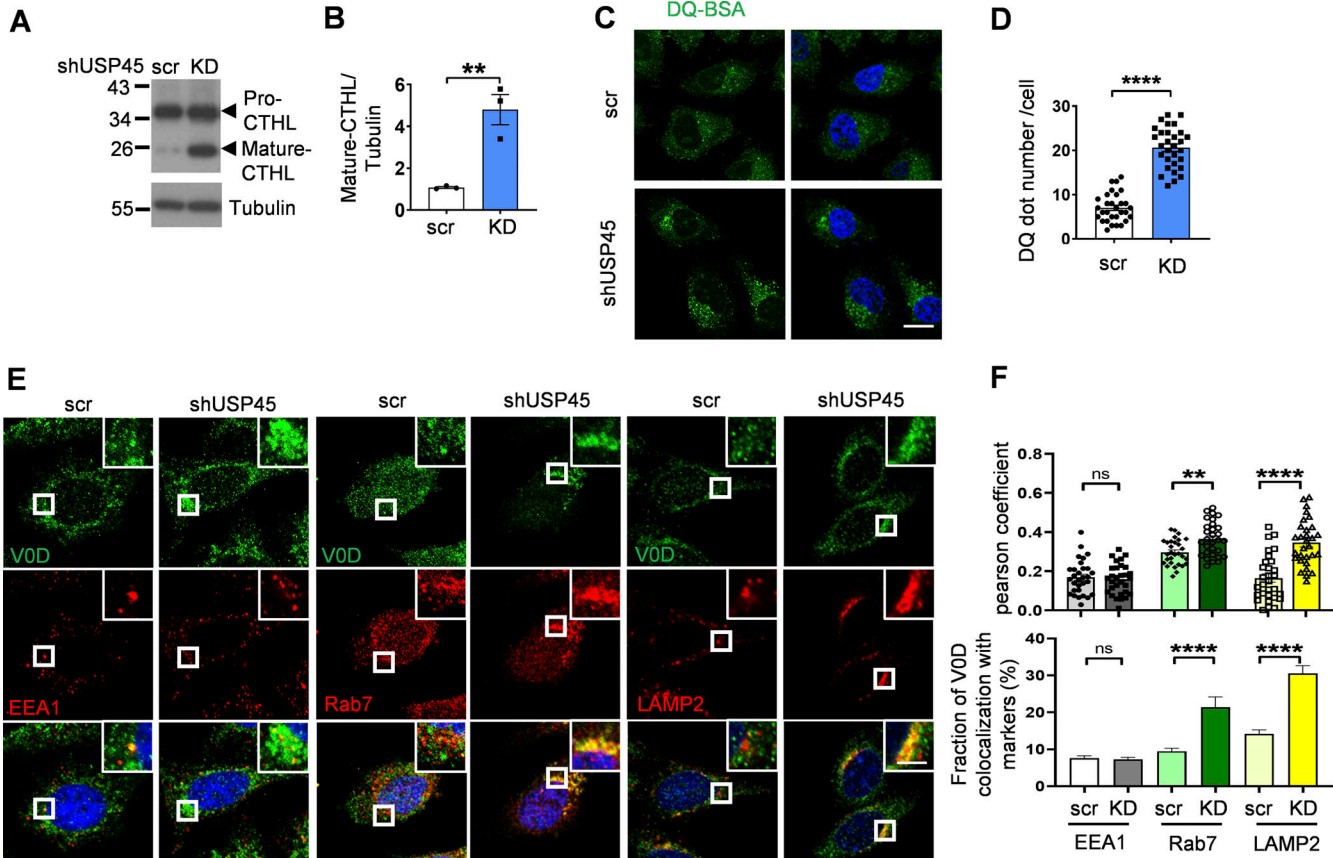

Figure 4. **USP45 negatively regulates lysosomal activity by altering V-ATPase endolysosomal trafficking. (A)** Western blot analysis of Cathepsin-L (CTHL) levels in control (scr) and *USP45* knockdown HeLa cells. **(B)** Quantification of mature CTHL expression normalized to tubulin in A. Data shown as mean ± SEM of three independent experiments. **(C)** Immunofluorescence analysis of DQ-BSA puncta in control or *USP45* knockdown HeLa cells. Scale bar, 20 µm. **(D)** Quantification of the number of DQ-BSA puncta in control and *USP45* knockdown cells treated as C; data are shown as mean ± SEM, n = 3, ≥ 30 cells. **(E)** Confocal microscopy analysis of colocalization of the V-ATPase subunit (V0D) and endolysosomal markers (EEA1, Rab7, and LAMP2) in control and *USP45* knockdown cells with indicated antibodies. The scale bars show 10 µm (original) and 4 µm (zoom-in). **(F)** Quantification of the colocalization of V0D and endolysosomal markers in E. Pearson's correlation coefficient and colocalization rate were analyzed by ImageJ. Data are shown as mean ± SEM, n = 3, ≥ 30 cells. Significance was determined using Student's t test (B, D, and F); *P < 0.05; **P < 0.01; ***P < 0.001; ****P < 0.0001; NS not significant. Source data are available for this figure: SourceData F4.

## USP45 interacts with and regulates the protein stability of Coronin 1B

Previous studies have indicated the involvement of the actin cytoskeleton and myosin motor proteins in the regulation of autophagy (Kruppa et al., 2016). Interestingly, in a global proteomic analysis of DUB-associated protein complexes, Sowa et al. reported that mammalian USP45 may interact with several actin and myosin regulators, including Coronin 1C (Coro1C), the non-muscle myosin heavy chain (MYH9 and MYH10), and the myosin regulatory light chain MRLC2 (Sowa et al., 2009). Recent research has suggested that the two widely distributed type I coronins, Coro1B and Coro1C, play a significant role in regulating branched actin turnover and cell motility (King et al., 2022; Striepen and Voeltz, 2022). However, their involvement in autophagy remains largely ambiguous. Interestingly, coimmunoprecipitation (co-IP) assays showed that USP45 bound to Coro1B but not Coro1C via its N-terminal seven-bladed β-propeller domain (Fig. 5, A and B; and Fig. S3, A and B). USP45 has been demonstrated to modulate the stability of Spindly and MYC proteins through deubiquitination (Conte et al., 2018; Tu et al., 2023). Next, we examined whether USP45 might act as a deubiquitinase toward Coro1B and regulate its protein levels. As shown in Fig. 5, C and D, the protein levels of Coro1B but not Coro1C were dramatically decreased in USP45-depleted cells, whereas MG132 treatment significantly inhibited Coro1B degradation (Fig. 5, E and F). Furthermore, the cycloheximide-chase assay revealed that USP45 depletion enhanced the protein degradation of Coro1B (Fig. 5, G and H). Since USP45 depletion did not affect *CORO1B* mRNA expression (Fig. S3 C), our results suggest that Coro1B might be a target substrate of USP45. The in vivo ubiquitination assay showed that USP45 depletion led to increased levels of ubiquitinated Coro1B (Fig. 5, I and J). Moreover, expression of USP45-WT, but not the catalytically inactive USP45 (C199A) mutant, dramatically decreased the ubiquitination of Coro1B under both mild (1% Triton) and denaturing (1% SDS) conditions (Fig. 5, K and L; and Fig. S3 D), suggesting that USP45 protease activity is required for the deubiquitination of Coro1B.

## USP45 depletion–induced F-actin patch formation is crucial for V-ATPase lysosome localization and lysosomal acidification

Coronins are conserved actin-binding proteins that modulate actin networks and have been implicated in many cellular processes (Gandhi and Goode, 2008). However, the precise role of coronins in autophagy regulation remains to be elucidated. Since USP45 mediates Coro1B deubiquitination and regulates its stability, we investigated whether USP45 might regulate actin dynamics by targeting Coro1B. Examination of endogenous F-actin with phalloidin staining revealed significantly increased patch-like actin structures in USP45-depleted cells compared with controls (Fig. 6, A and B). The F-actin patches induced by USP45 knockdown highly colocalized with the late endosomal marker Rab7 and the lysosomal marker LAMP1, but not with the early endosomal marker EEA1 (Fig. 6, C and D), suggesting a role for cytosolic F-actin patches in regulating endolysosomal activity. Indeed, the F-actin patches induced by USP45 depletion showed high colocalization with LysoTracker-positive acidic lysosomes

(Fig. S4 A). Treatment with the actin polymerization inhibitor latrunculin A (LatA) resulted in a rapid reduction of LysoTracker puncta staining in USP45-depleted cells (Fig. S4, A and B). Notably, *USP45* knockdown cells quickly regained acidic lysosomes that colocalized with F-actin patches following LatA washout and replenishment with fresh media (Fig. S4, A and B). We next examined whether the F-actin patches induced by USP45 depletion were required for the translocalization of V-ATPase to lysosomes. As shown in Fig 6, E and F, the colocalization of V-ATPase and lysosomes was dramatically reduced in the presence of actin polymerization inhibitor LatA and Arp2/3 inhibitor CK666 in *USP45* knockdown cells. The Arp2/3 complex cooperates with the nucleation-promoting factors such as the Wiskott–Aldrich Syndrome Protein (WASP) family proteins to regulate actin filament assembly (Padrick et al., 2011; Rohatgi et al., 1999). It has been shown that WASH (a member of the WASP family)-mediated actin polymerization is crucial for lysosomal neutralization and recycling of V-ATPases in *Dictyostelium* (Carnell et al., 2011). Consistently, we observed a dramatic increase in the colocalization of V-ATPase and the lysosomal marker LAMP1 upon depletion of WASH in both control and USP45-depleted cells (Fig. 6, G and H). In contrast, the knockdown of N-WASP, a ubiquitously expressed homolog of WASP, caused a significant reduction of V-ATPase on lysosomes in USP45-depleted cells compared with controls (Fig. 6, G and H). Taken together, our results suggest that Arp2/3 and N-WASP-, but not WASH-, mediated actin polymerization plays a critical role in regulating V-ATPase translocation to lysosomes in USP45-depleted cells.

## Coro1B inhibits autophagy and lysosomal function

It has been reported that Coro1B regulates actin dynamics by coordinating the Arp2/3 complex and the actin-depolymerizing factor/cofilin activities (Cai et al., 2007). However, the molecular function of Coro1B in autophagy/lysosome remains unclear. We first checked whether the knockdown of Coro, the type I coronin homolog in *Drosophila*, might affect autophagy. The tandem-Atg8a reporter assay revealed a significant increase of mCherry⁺ GFP- autolysosomes in Coro-depleted larval fat body cells compared with controls (Fig. 7, A and B). Moreover, genetic analysis showed that depletion of Coro markedly suppressed dUsp45-induced autolysosome enlargement in wandering wL3 larval fat body cells (Fig. 7, C and D). Similarly, knockdown of *twinstar (tsr)*, the *Drosophila cofilin* gene, or overexpression of the dominant negative *tsr^DN^* mutant, but not the constitutively active *tsr^CA^* mutant, dramatically suppressed the number of aberrantly enlarged autolysosomes in *dUsp45* overexpressing larval fat body cells (Fig. 7, C and D). Our epistasis tests suggested that Coro and Tsr act downstream of dUsp45 in the regulation of autolysosomal function in *Drosophila*.

We next checked whether mammalian Coro1B also plays a similar role in the regulation of lysosomal function. As shown in Fig. S5, A–D, Coro1B depletion resulted in elevated levels of LysoTracker Red staining and DQ-BSA puncta. Conversely, the ectopic expression of Coro1B caused a reduction in the LysoTracker dot number in control and *USP45* knockdown cells (Fig. S5, E and F). These findings suggested that Coro1B negatively

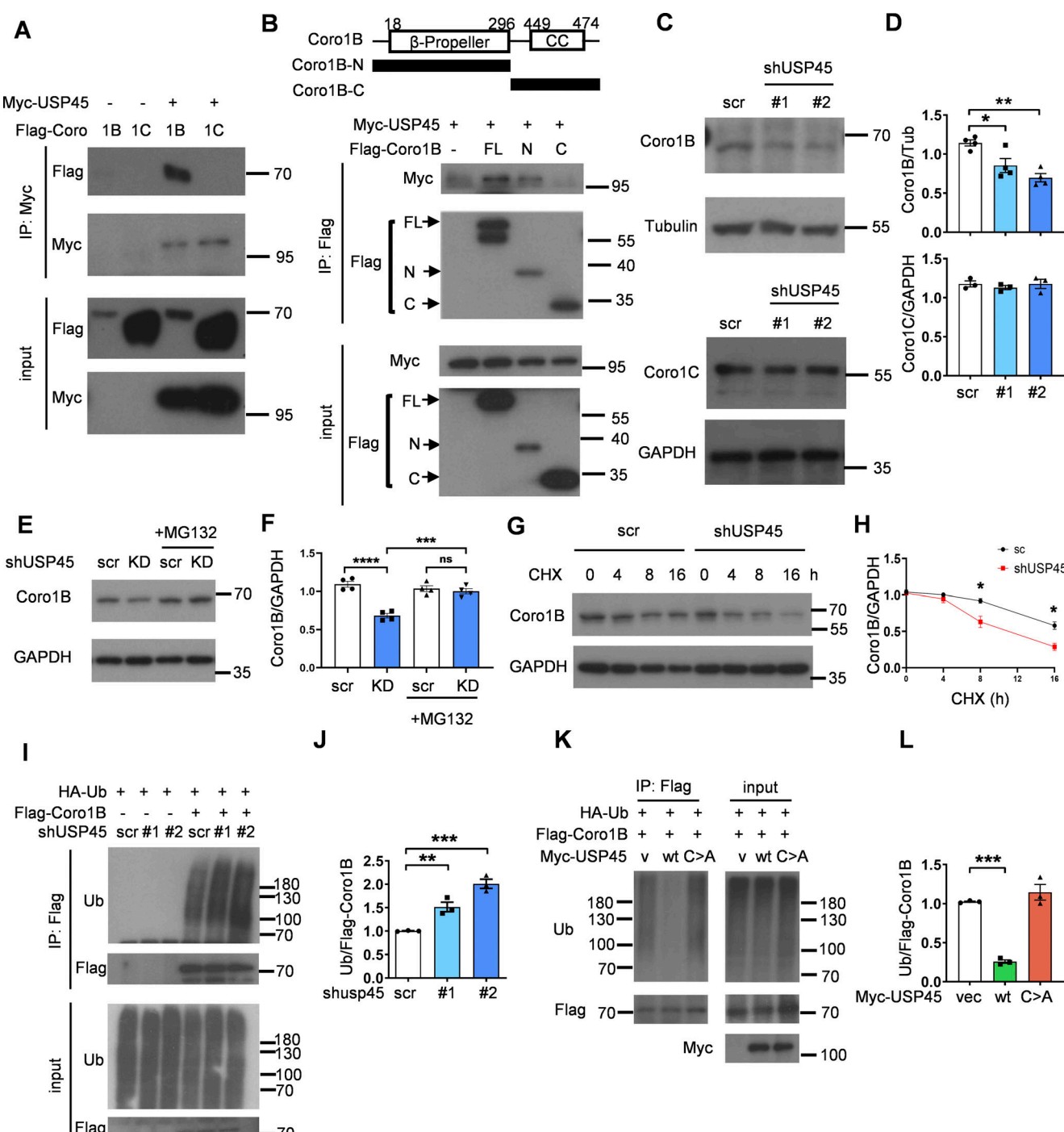

Figure 5. **USP45 interacts with and deubiquitinates Coro1B. (A)** Coimmunoprecipitation (Co-IP) analysis of the interaction between Myc-USP45 and Flag-Coro1B or Flag-Coro1C in HEK293T cells. Co-IP experiments were performed using an anti-Myc antibody, followed by a Western blot analysis with the indicated antibodies. **(B)** (top). Schematic presentation of domain structures and deletion mutants of Coro1B. β-propeller, β-propeller domain; CC, coiled-coil domain. (bottom) Coimmunoprecipitation assays to map the interaction regions between USP45 and Coro1B. **(C)** Western blot analysis of the expression levels of Coro1B and Coro1C with indicated antibodies in control and *USP45* knockdown HeLa cells. **(D)** Quantification of Coro1B and Coro1C expression normalized to tubulin and GAPDH, respectively. Data are shown as mean ± SEM of at least three independent experiments. **(E)** Western blot analysis of the expression levels of Coro1B in control (scr) and *USP45* knockdown HeLa cells with or without treatment of proteasome inhibitor MG132 (5 μM, 4 h). **(F)** Quantification of Coro1B expression normalized to GAPDH in E. Data are shown as mean ± SEM of four independent experiments. **(G)** Cycloheximide (CHX) chase analysis of Coro1B expression in control and *USP45* knockdown HeLa cells treated with CHX for the indicated times. **(H)** Quantification of Coro1B levels normalized to GAPDH at different time points after CHX treatment, as indicated in G. Data shown as mean ± SEM of four independent experiments. **(I)** Immunoprecipitation analysis for Coro1B ubiquitination in control or *USP45* knockdown HEK293T cells. **(J)** Quantification of ubiquitin (Ub) levels normalized to Coro1B in I. Data are shown as mean ± SEM of three independent experiments. **(K)** Immunoprecipitation analysis of Coro1B ubiquitination in cells expressing Flag-Coro1B, Myc-tagged USP45-WT or the catalytically inactive USP45-C199A (C > A) mutant. **(L)** Quantification of Ub levels normalized to Coro1B in K. Data shown as mean ± SEM of three independent experiments. Significance was determined using one-way ANOVA and Dunnett's (D, J, and L) or Tukey's (F) multiple comparisons test, and Student's *t* test (H); *P < 0.05; **P < 0.01; ***P < 0.001; ****P < 0.0001; NS not significant. Source data are available for this figure: SourceData F5.

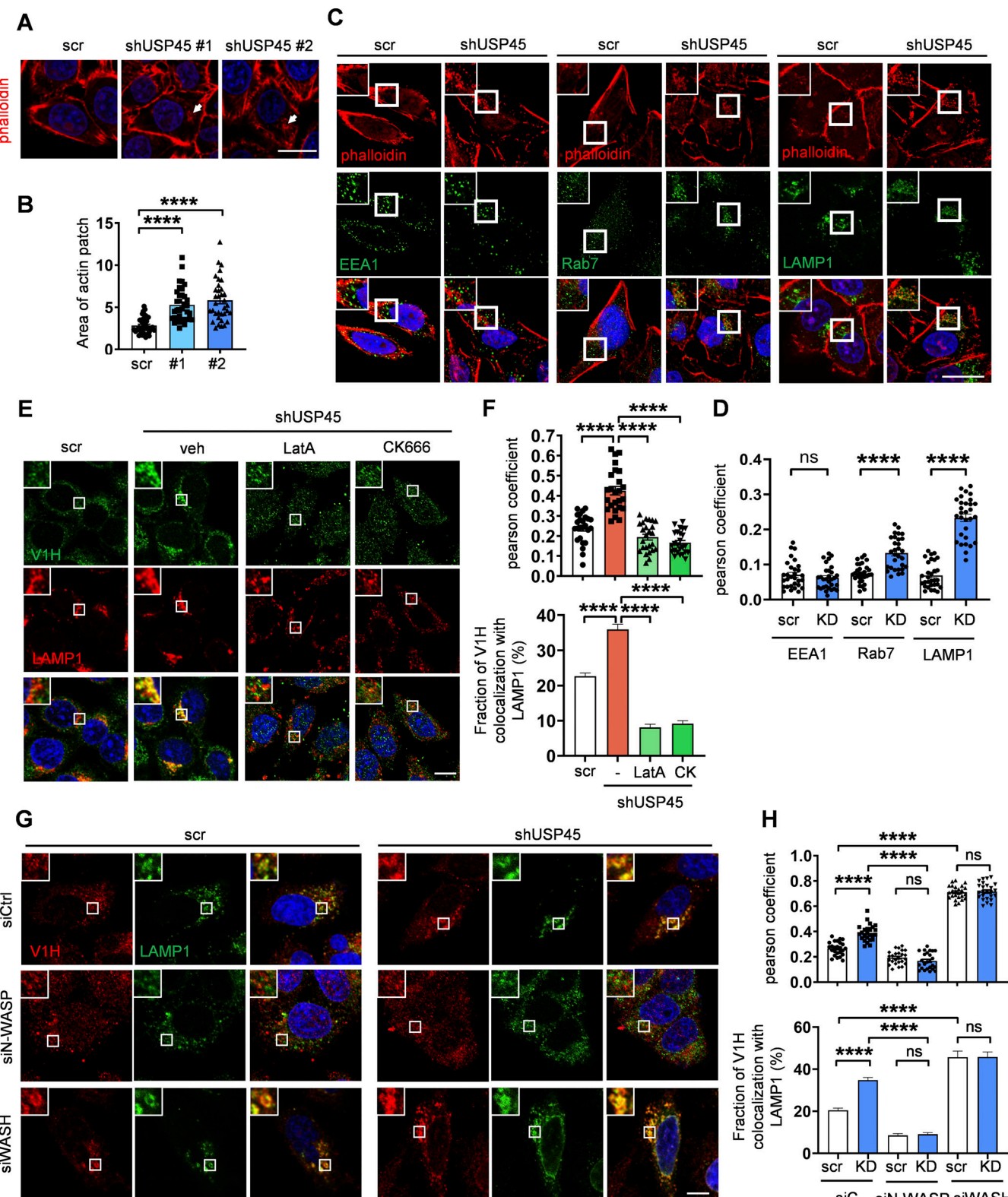

Figure 6. **USP45 depletion–induced F-actin patch formation is required for V-ATPase translocation to lysosomes. (A)** Confocal microscopy analysis of F-actin structures stained with phalloidin in control (scr) and *USP45* knockdown HeLa cells. Arrows indicate cytoplasmic F-actin patches. Scale bar, 20 μm. **(B)** Quantification of the area of cytoplasmic F-actin patches in A; data shown as mean ± SEM, *n* = 3, ≥35 cells. **(C)** Immunofluorescence analysis of co-localization of the F-actin patches and endolysosomal markers (EEA1, Rab7, and LAMP1) in control and *USP45* knockdown cells with indicated antibodies. The scale bars showed 20 μm (original) and insets with 1.5x zoom. **(D)** Quantification of the colocalization of F-actin patches and endolysosomal markers in C. Pearson's correlation coefficient was analyzed by ImageJ. Data are shown as mean ± SEM, *n* = 3, ≥ 30 cells. **(E)** Immunofluorescence analysis of colocalization of V-ATPase (V1H) and lysosome (LAMP1) in control and *USP45* knockdown HeLa cells treated with LatA (200 nM) or Arp2/3 inhibitor CK666 (200 μM) for 2 h.

Scale bar, 10 µm (original) and insets, 2.5x zoom. **(F)** Quantification of the colocalization of V1H and LAMP1 in E. Pearson's correlation coefficient and co-localization rate were analyzed by ImageJ. Data are shown as mean ± SEM, $n = 3$, ≥ 25 cells. **(G)** Immunofluorescence analysis of colocalization of V-ATPase (V1H) and lysosome (LAMP1) in control and *USP45* knockdown cells transfected with control (siCtrl) siRNA or siRNA targeting WASP family genes (siN-WASP and siWASH). Scale bar, 10 µm (original) and insets, 2.5x zoom. **(H)** Quantification of the colocalization of V1H and LAMP1 in G. Data are shown as mean ± SEM, $n = 3$, ≥ 25 cells. Significance was determined using one-way ANOVA and Dunnett's (B) or Tukey's (F and H) multiple comparisons test, and Student's $t$ test (D); ***P < 0.001; ****P < 0.0001; NS not significant.

regulated lysosomal acidity and proteolytic activity. Given our earlier observation that USP45 regulated V-ATPase translocation to lysosomes by modulating actin dynamics, we examined whether Coro1B also regulates V-ATPase translocation to lysosomes. Immunofluorescence analysis revealed a dramatic increase in V-ATPase localization at lysosomes in Coro1B-depleted cells compared with controls (Fig. 7, E and F). Treatment of Coro1B knockdown cells with the actin polymerization inhibitor LatA or the Arp2/3 inhibitor CK666 resulted in a significant reduction of V-ATPase localization to lysosomes (Fig. 7, G and H). The increased levels of lysosomal V-ATPase in Coro1B knockdown cells were further confirmed by LysoIP analysis (Fig. S5, G and H). Moreover, the knockdown of N-WASP but not WASH diminished the Coro1B depletion–induced lysosomal localization of V-ATPases (Fig. S5, I and J). Together, our observations indicate a negative role of Coro1B in the regulation of lysosomal function in *Drosophila* and mammalian cells.

### Loss of *dUsp45* extends lifespan and ameliorates polyQ-induced eye degeneration in *Drosophila*

There is growing evidence that a decline in autophagy and lysosomal activity plays a major role in a variety of aging-related diseases, such as neurodegenerative disorders (Aman et al., 2021; Barbosa et al., 2019). Because USP45 plays a critical role in the regulation of autophagy and lysosomal function, we investigated the physiological function of dUsp45 in *Drosophila* aging and longevity. As shown in Fig. 8, A and B, we observed a significant reduction in mature cathepsin-L protein levels in aged flies compared with young flies, suggesting a decline in lysosomal activity during aging. Intriguingly, we also observed a significant increase in the mRNA levels of *dUsp45* in aging flies (Fig. 8, C and D), suggesting an inverse correlation between *dUsp45* expression and lysosomal function during aging in *Drosophila*. To further determine the role of dUsp45 in aging and longevity, we examined the lifespan of *dUsp45* mutant flies, including the *dUsp45* homozygote P-element insertion mutant (*dUsp45^EY19835^/dUsp45^EY19835^*) and the *dUsp45* trans-heterozygote mutant (*dUsp45^EY19835^/dUsp45^B^*) (Fig. 8 E). As shown in Fig. 8 F, both *dUsp45* mutants significantly increased mean lifespan by 15–17% (P < 0.001) compared with w1118 control flies and by 10–12% (P < 0.001) compared with *dUsp45* heterozygous control flies. PolyQ diseases are neurodegenerative diseases caused by an abnormal expansion of glutamine repeats in disease-causative proteins. The *Drosophila* compound eye consists of ~800 ommatidia, each with seven of the eight photoreceptor neurons found in coronal sections (Katz and Minke, 2009). It has been reported that the expanded polyQ tracts induce neuronal degeneration in the *Drosophila* eye (Xu et al., 2015). We next checked whether ablation of dUsp45 or Coro could mitigate the

polyQ-induced toxicity in *Drosophila*. Ectopic expression of expanded polyQ proteins (72Q) in developing eyes with the *GMR-Gal4* driver caused a disruption of regular photoreceptor array and loss of rhabdomeres in an age-dependent manner (Fig. 8, G and J). Notably, knockdown of *dUsp45* or *Coro* dramatically rescued the 72Q-induced degeneration defects in aged flies (Fig. 8, G and J), suggesting that downregulation of dUsp45 and coronin could protect cells from polyQ-induced toxicity.

## Discussion

Lysosomes are acidic organelles that function as intracellular degradation and signaling hubs essential for maintaining cellular homeostasis (Ballabio and Bonifacino, 2020). Lysosomal acidification impairment has been associated with aging and neurodegenerative diseases. In this study, we identified dUsp45/USP45 deubiquitinase as a negative regulator of autophagy and lysosomal function. Our results in both *Drosophila* and mammalian cells showed that ablation of dUsp45/USP45 leads to increased autophagosome formation and enhanced autophagic flux. Depletion of USP45 resulted in reduced mTOR activity, enhanced TFEB nuclear translocation, and increased expression of autophagy initiation proteins, suggesting that USP45 plays a critical role in modulating autophagy through the regulation of mTOR activity. Additionally, we found that knockdown of *dUsp45/USP45* promotes autolysosome formation and enhances lysosomal activity, whereas overexpression of *dUsp45* leads to the formation of enlarged autolysosomes with reduced acidity, suggesting a negative role for dUsp45/USP45 in regulating lysosomal acidification. Further, we showed that depletion of *dUsp45/USP45* promotes V-ATPase localization to autolysosomes, thereby enhancing lysosomal acidification and autophagic–lysosomal activity. Moreover, we demonstrated that USP45 regulates V-ATPase translocation by modulating Coro1B-mediated actin dynamics. Taken together, our findings reveal USP45's role in controlling lysosomal activity through the modulation of the actin cytoskeleton network, suggesting that it is a potential target for treating age-related diseases, including neurodegenerative disorders (Fig. 8 K).

The lysosome function requires the acidification of the lysosomal lumen to a lower pH, which is mediated by a number of ion channels and most notably by the V-ATPase. The V-ATPase is a large multisubunit protein complex composed of a cytoplasmic ATP-hydrolyzing V1 domain and a membrane-embedded V0 domain responsible for proton pumping (Collins and Forgac, 2020). The reversible dissociation of the V1 and V0 domains, known as regulated assembly, plays a critical role in regulating the activity of V-ATPases. In addition to regulated assembly, the targeting of V-ATPases to distinct cellular

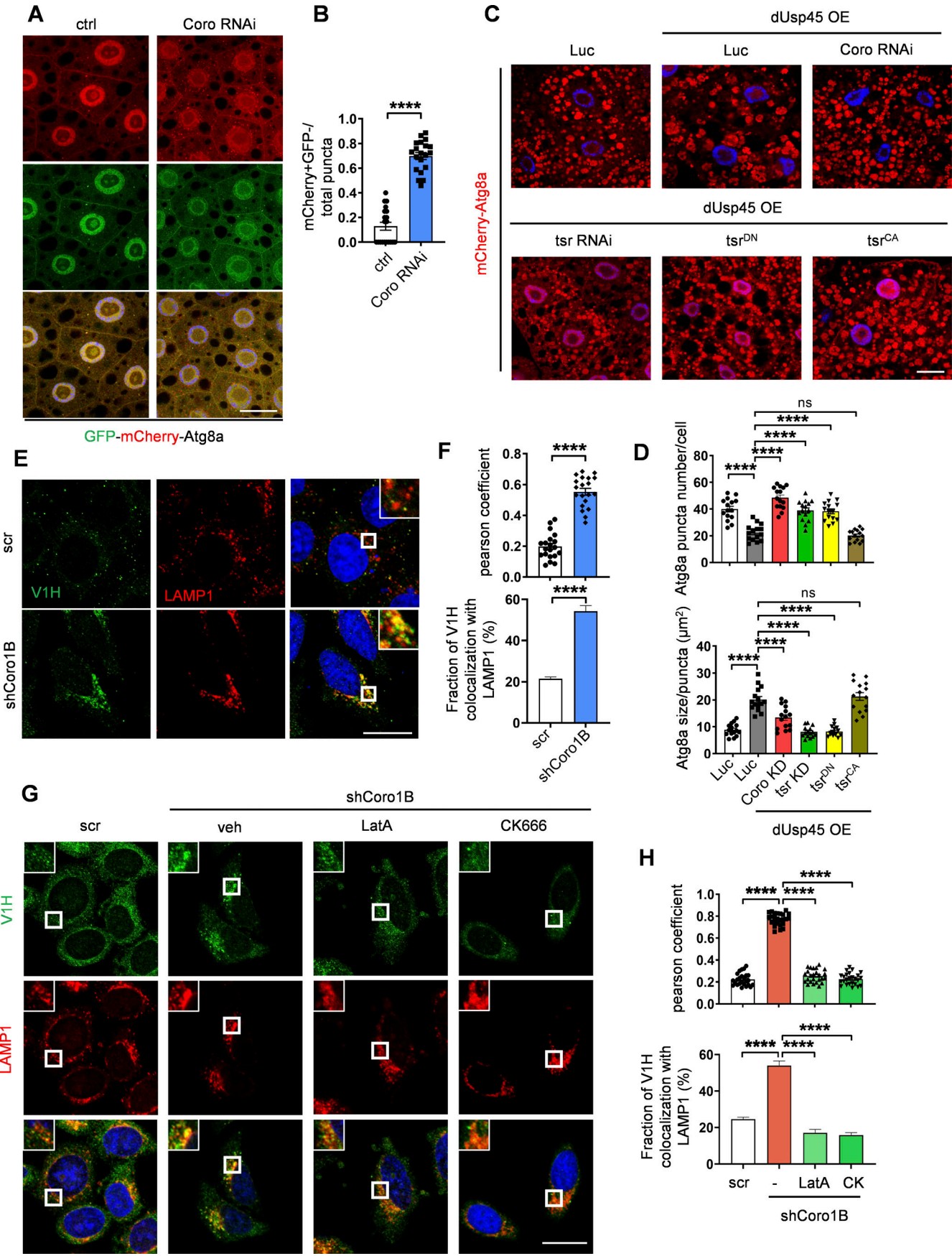

Figure 7.   **Depletion of Coro1B promotes autophagy and V-ATPase lysosomal localization. (A)** Confocal images of late L2 larval fat body cells expressing GFP-mCherry-Atg8a in control (Ctrl) and *Coro* knockdown larvae. Scale bar, 20 μm. **(B)** Quantification of the ratio of autolysosomes (GFP⁻ mCherry⁺) to total Atg8a puncta per cell in A; data are shown as mean ± S EM, *n* = 3, ≥ 20 cells. **(C)** Confocal images of wL3 larval fat body cells from flies expressing mCherry-Atg8a and indicated transgenes driven by *Cg-Gal4*. Scale bar, 20 μm. **(D)** Quantification of Atg8a puncta number and size shown in C. Data are shown as mean ± SEM, *n* = 3, ≥ 15 cells. **(E)** Immunofluorescence analysis of colocalization of V-ATPase (V1H) and lysosome (LAMP1) in control and *Coro1B* knockdown HeLa cells. Scale bar, 20 μm (original) and insets, 3x zoom. **(F)** Quantification of the colocalization of V1H and LAMP1 in E. Pearson's correlation coefficient and co-localization rate were analyzed by ImageJ. Data are shown as mean ± SEM, *n* = 3, ≥ 20 cells. **(G)** Immunofluorescence analysis of colocalization of V-ATPase (V1H) and lysosome (LAMP1) in control and *Coro1B* knockdown cells treated with LatA (200 nM) or Arp2/3 inhibitor CK666 (200 μM) for 2 h. Scale bar, 20 μm (original) and insets, 2x zoom. **(H)** Quantification of the colocalization of V1H and LAMP1 in G. Data are shown as mean ± SEM, *n* = 3, ≥ 25 cells. Significance was determined using one-way ANOVA and Tukey's multiple comparisons test (D and H), and Student's *t* test (B and F); ****P < 0.0001; NS not significant.

membranes is crucial for regulating their function. The membrane localization of V-ATPases is regulated by isoforms of the V0 domain a-subunit (a1–a4) (McGuire et al., 2016). It has been shown that a1 and a2 mainly target V-ATPases to intracellular membranes, whereas a3 and a4 are localized to the plasma membrane of specialized cells (Hurtado-Lorenzo et al., 2006; Morel et al., 2003; Toyomura et al., 2003). Our data revealed that the assembly of V-ATPase was not affected in USP45-depleted cells. Instead, USP45 regulates lysosomal acidification by modulating the intracellular localization of v-ATPase. We found that knockdown of USP45 dramatically enhanced the localization of V-ATPase to late endosomes and lysosomes, thereby enhancing the number of acidified lysosomes and lysosomal function.

Accumulating evidence has highlighted the pivotal roles of ubiquitination and deubiquitination in governing endocytic trafficking (Piper et al., 2014). However, their implications in orchestrating actin filament organization and autolysosomal function remain enigmatic. Several lines of evidence indicate that the USP45 deubiquitinase regulates autophagy activity and lysosomal function by modulating actin dynamics. Firstly, USP45 interacts with and deubiquitinates the actin-associated protein Coro1B, thereby stabilizing its protein levels. Moreover, it has been reported that Coro1B coordinates the Arp2/3 complex and cofilin to regulate branched actin networks (Cai et al., 2007), which are involved in the regulation of endocytic trafficking and the maintenance of endosomal morphology. Our findings reveal that knocking down *USP45* leads to the formation of cytoplasmic actin patches, closely associated with late endosomes and lysosomes. Additionally, inhibiting actin polymerization via LatA or CK666 impedes the V-ATPase lysosomal translocation and autolysosomal formation induced by *USP45* or *Coro1B* knockdown. Notably, autolysosomal defects induced by dUsp45 overexpression can be mitigated by fly *coronin* or *twinstar (tsr)* knockdown, or by expressing a dominant negative tsr variant. While previous studies have demonstrated that USP45 can catalyze the deubiquitination of a variety of ubiquitin chains, including K6-, K48-, and K63-linked polyubiquitination (Conte et al., 2018), the precise mechanism governing Coro1B stability regulation by USP45 remains elusive.

Recent studies have indicated the involvement of the actin cytoskeleton in the regulation of V-ATPase translocation. The V-ATPase V1B and V1C subunits were found to interact with the actin cytoskeleton and are necessary for the trafficking of V-ATPase to the apical membrane of epithelial cells of the tobacco hornworm (Vitavska et al., 2003). Moreover, the actin nucleation factor WASH-mediated actin polymerization has been reported to regulate lysosome neutralization and V-ATPase recycling in *Dictyostelium* (Carnell et al., 2011; Park et al., 2013). Intriguingly, we found that N-WASP-, but not WASH-, mediated F-actin nucleation is required for the translocation of V-ATPase to lysosomes. Our findings suggest that N-WASP and WASH regulate distinct branched actin networks, with N-WASP controlling V-ATPase lysosomal translocation and WASH potentially involved in the retrieval of V-ATPase from lysosomes. Given that a large fraction of USP45 is localized to endosomes and lysosomes, it is likely that the USP45–Coro1B axis imposes strict spatial and temporal control on N-WASP activity within the endo-lysosome system, thereby regulating V-ATPase localization and lysosomal function. The actin cytoskeleton has pleiotropic effects on a wide range of cellular processes, and our proposed mechanism represents one plausible model. However, further studies are needed to fully elucidate the role of the actin cytoskeleton in V-ATPase trafficking and lysosomal function.

A growing body of evidence indicates that autophagy plays a crucial role in removing misfolded or unfolded protein aggregates during aging (Aman et al., 2021). Compromised autophagic activity often leads to premature aging and a shortened lifespan. Our findings revealed that dUsp45/USP45 acts as a negative regulator of autophagy and lysosomal function. The expression of *dUsp45* was dramatically increased in aged flies. Importantly, loss of *dUsp45* leads to an extended lifespan, suggesting the role of dUsp45 in regulating longevity and healthy aging. Moreover, we observed that ablation of dUsp45 or coronin could mitigate polyQ-associated neurodegeneration in a *Drosophila* model. Interestingly, several recent reports have implicated the involvement of USP45 in cancer progression (Li et al., 2022; Tu et al., 2023). Future studies will focus on validating USP45's role in human diseases.

## Materials and methods
### *Drosophila* stocks, genetics, and treatment
Flies were reared at 25°C following standard procedures. The fly strains used in this study were listed below: *GMR-Gal4, Cg-Gal4* (BL7011), *UASp-mCherry-GFP-Atg8a* (BL37749), *UAS-Luciferase* (BL31603), *UAS-Dcr2* (BL24650, BL24651), *dUsp45^{EYI9835}* (BL22338), *dUsp45^B* (BL57080), *VhaSFD* (BL6840) and *Vha13* (BL50828), *UAS-tsr^{RNAi}* (BL65055), *UAS-tsr^{S3A}* (BL9237), *UAS-*

Figure 8. **Loss of *dUsp45* extends lifespan and mitigates polyQ-induced toxicity in *Drosophila*. (A)** Western blot analysis of cathepsin-L (CTHL) levels in the heads of adult flies at different ages. **(B)** Quantification of CTHL levels normalized to Tubulin in A. Data are shown as mean ± SEM of three independent

experiments. **(C)** RT-PCR analysis of *dUsp45* mRNA levels in the heads of adult flies at different ages. **(D)** Quantification of *dUsp45* mRNA levels normalized to *Actin*. Data are shown as mean ± SEM of three independent experiments. **(E)** RT-PCR analysis of *dUsp45* mRNA levels in control (*w1118* and *dUsp45*[EY/+]) and *dUsp45* mutant (*dUsp45*[EY/EY] and *dUsp45*[EY/B]) flies. Numbers below lanes indicate the relative ratio of *dUsp45/Actin*. **(F)** Loss of *dUsp45* (*dUsp45*[EY/EY] and *dUsp45*[EY/B]) extends lifespan compared with controls (*w1118* and *dUsp45*[EY/+]). $N \geq 582$ flies of each genotype. **(G)** Confocal images of phalloidin stained retinae from young (1 day) and aged (70 days) adult flies expressing GFP (control), or mutant (*72Q*) *Htt* fragment together with *GFP* or *dUsp45*[RNAi] under control of the eye-specific driver *GMR-Gal4*. Flies were reared in a light/dark (LD 12L:12D) cycle incubator. Scale bar, 10 µm. **(H)** Quantification of intact rhabdomere numbers in flies described in G. Data shown as mean ± SEM in *n* = 3, ≥30 ommatidea per genotype. **(I)** Confocal images of phalloidin stained retinae from young (1 day) and aged (40 days) adult flies expressing GFP (control), or mutant (*72Q*) *Htt* fragment together with *GFP* or *Coro*[RNAi] under control of the eye-specific driver *GMR-Gal4*. Flies were reared in a light/dark (LD 12L:12D) cycle incubator. Scale bar, 10 µm. **(J)** Quantification of intact rhabdomere numbers in flies described in I. Data shown as mean ± SEM in *n* = 3, ≥20 ommatidea per genotype. **(K)** A schematic diagram illustrating the role of USP45 in regulating autophagy, lysosome function, and V-ATPase lysosomal localization through the modulation of actin structures via Coro1B. Significance was determined using one-way ANOVA and Tukey's multiple comparisons test (B, D, H, and J), and log-rank test (F); *$P < 0.05$; **$P < 0.01$; ***$P < 0.001$; ****$P < 0.0001$; NS not significant. Source data are available for this figure: SourceData F8.

*tsr*[S3E] (BL9239), and *UAS-HTT.72Q.Emerald* (BL58361) were obtained from the Bloomington Stock Center. *UAS-dUsp45*[RNAi#1] (v41976), *UAS-dUsp45*[RNAi#2] (v110286), and *UAS-coro*[RNAi] (v44672) were obtained from the VDRC stock center. *yw hsflp; r4-mCherry-Atg8a Act>CD2>Gal4 UAS-GFPnls* and *UAS-mCherry-Atg8a* were kind gifts from Thomas Neufeld (Chang and Neufeld, 2009). The *UAS-Flag-dUsp45* was constructed by subcloning dUsp45 cDNA into a pUAST vector. The *UAS-dUsp45*[C3I5A] mutant was generated by site-directed mutagenesis. For the CQ treatment, a particular stage of larvae was collected and incubated in normal food with chloroquine (10 mg/ml, Sigma-Aldrich) for 6 or 16 h. For the lifespan assay, 20 flies were reared in each tube under a 12 h light/12 h dark cycle. Flies were transferred to fresh food every 3 days until all flies died.

### Plasmids and shRNA

The Myc-USP45 (WT and C199A mutant) was obtained from MRC PPU. The Flag-Coro1B, Flag-Coro1C, and Coro1B truncation plasmids were constructed by PCR amplification and subcloned into pcDNA3.1 (V790-20; Invitrogen). The pLJC5-TMEM192-2xFlag (#102929) and pLJC5-TMEM192-3xHA (#102930), pRK5-HA-Ubiquitin-WT (#17608) plasmids were acquired from Addgene. The GFP-TFEB plasmid was a gift from Wei Yuan Yang at Academia Sinica. The lentiviral shRNA clones were obtained from the National RNAi Core Facility of Academia Sinica. The target sequences of shRNA clones are shUSP45#1 (ID: TRCN0000253807, 5′-TCAGAAGGCACCTCGATTTAA-3′), shUSP45#2 (ID: TRCN0000253809, 5′-ATCTGAGCACATGGATTATAT-3′), shCoro1B#1 (ID: TRCN0000116422, 5′-CGCCCAGCTTTCCTCACTGTT-3′), and shCoro1B#2 (ID: TRCN0000116426, 5′-CGTGGTACTCATCTGGAATGT-3′). shLacZ (ID: TRCN0000072224, 5′-CGCGATCGTAATCACCCGAGT-3′) was used as a control. The siRNA for siN-WASP or siWASH contained a mixture of four siRNAs covering the target gene sequence (SMARTpool), obtained from Dharmacon. siN-WASP (SMARTpool, L-006444-00-0005): 5′-CAGCAGAUCGGAACUGUAU-3′, 5′-UAGAGAGGGUGCUCAGCUA-3′, 5′-CGUGUUGCUUGUCUUGUUA-3′, 5′-CCAGAAAUCACAACAAAUA-3′; siWASH (SMARTpool, L-190043-00-0005): 5′-AGCAGGUCCCAGAGAACUA-3′, 5′-AGACCUAUGCCGUGCCCUU-3′, 5′-GUGCAGGCCAUUGGAGAGA-3′, 5′-AGACCUACAAGAUGGGGUA-3′. The non-targeting siRNA used was SMARTpool D-001810-10-05: 5′-UGGUUUACAUGUCGACUAA-3′, 5′-UGGUUUACAUGUUGUGUGA-3′, 5′-UGGUUUACAUGUUUUCUGA-3′, 5′-UGGUUUACAUGUUUUCCUA-3′.

### Cell cultures and treatments

HeLa and HEK293T cells (gifts from Ruey-Hwa Chen; Academia Sinica) were cultured in DMEM medium (Invitrogen) supplemented with 10% fetal bovine serum (FBS) and 1% antibiotics (penicillin-streptomycin) at 37°C. Transfection was conducted using PolyJet reagent (SignaGen Laboratories). Plasmids and PolyJet were added to a serum-free medium at a ratio of 1:2 and mixed for 15 min. The mixture was then added to the medium and incubated for 24–48 h before further experiments. To establish stable knockdown cell lines, shRNA plasmids were first mixed with Δ8.9 plasmids and vesicular stomatitis virus G protein (VSVG), followed by the transfection procedure mentioned above. The mixture was added to HEK293T cells for 24 h. The medium was then replaced with fresh medium and incubated for an additional 24–48 h. Virus-containing medium was collected, supplemented with 8 ng/ml polybrene, and added to the cells. 24 h after infection, cells were selected with a medium containing 2 µg/ml puromycin. After two to three rounds of selection, the target genes were stably depleted in surviving cells. In siRNA knockdown experiments, siRNA (10 µM) and Lipofectamine RNAiMAX Transfection Reagent (Invitrogen) were diluted in Opti-MEM medium (Gibco) at a ratio of 1:3 and mixed. The mixture was then incubated at room temperature for 5 min before being added to the cells for a 4 h incubation period. In the cycloheximide (CHX) chase assay, cells were treated with cycloheximide (100 µg/ml; Sigma-Aldrich) for the indicated times. For BafA1 (100 nM; Calbiochem) and MG132 treatment (5 µM; Sigma-Aldrich), the medium was replaced with fresh medium containing the specific chemicals and incubated for 4 h. For LatA (200 nM; Cayman) and CK666 treatment (200 µM; Sigma-Aldrich), cells were treated for 2 h. In the washout experiment, the medium was replaced with fresh DMEM following LatA treatment. At the specified time point, cells were washed with PBS and then fixed with 4% paraformaldehyde for further immunofluorescent analysis.

### Immunofluorescence

Fly larval fat bodies and adult eyes were fixed with 4% paraformaldehyde for 30 min at room temperature. Tissues were permeabilized with 0.3% Triton X-100 (Sigma-Aldrich)–PBS and blocked with 5% normal goat serum (Gibco) in 0.1% Triton-PBS. Samples were then incubated with the indicated primary antibodies at 4°C overnight. Following incubation, samples were washed with PBS and PBS containing 0.2% Triton X-100,

followed by incubation with secondary antibodies for 2 h at room temperature. Afterward, samples were washed three times with PBS, stained with DAPI, and mounted. Fly images were acquired using an Olympus FV3000 Confocal Microscopy with a 60×/1.40 oil objective. To quantify the size of Atg8a-positive puncta in the larval fat body, the threshold of Atg8a signals in secondary and wandering wL3 larvae was set above 0.05 and 0.3 μm, respectively, to exclude background signals using ImageJ 1.54J.

For immunofluorescence analysis of mammalian cells, cells at 70–80% confluence were gently washed three times with PBS and fixed with 4% paraformaldehyde at room temperature for 20 min or with methanol for 5 min at –20°C. After fixation, cells were permeabilized with 0.1% Triton X-100 in PBS containing 5% goat serum for 1 h at room temperature and then incubated with primary antibodies in PBS containing 2% goat serum at 4°C overnight. Cells were washed three times with PBS at room temperature for a total of 30 min and then incubated with secondary antibodies for at least 1 h. Nuclei were stained with DAPI and then the cells were mounted in mounting solution. Images were captured using an Olympus FV3000 Confocal Microscopy with a 60×/1.40 oil objective (UPlanSApo). The actin structure was stained with TRITC-labeled phalloidin (Invitrogen) for 2 h at room temperature. For LysoTracker Red (LTK, 1:1,000; Invitrogen) or DQ-BSA (10 μg/ml; Invitrogen) staining, they were added to the medium for 15 min and 6 h, respectively. Cells were then analyzed by confocal microscopy immediately after LTK and DQ-BSA treatment. The secondary antibodies used for immunofluorescence were as follows: Goat anti-rabbit IgG (H+L), Alexa Fluor 488 (cat# A11008), Alexa Fluor 598 (cat# A11011), Alexa Fluor 633 (cat# A21070); and Goat anti-mouse IgG (H+L), Alexa Fluor 488 (cat# A11001), Alexa Fluor 598 (cat# A11004), all from Invitrogen.

For colocalization analysis, the green (Ex 488 nm) and red (Ex 568 nm) images were examined by ImageJ. After the background subtraction, the colocalization results were output by the ImageJ plugin, JACoP.

## Antibodies

The following antibodies were used in this study: anti-LC3b (1:200, cat#4108), anti-LAMP1 (1:500, cat#9091), anti-EEA1 (1:200, Cat#3288), anti-Calnexin (1:1,000, cat#2679), anti-GAPDH (1:2,000, cat#5174), anti-ULK1 (1:1,000, cat#8054), anti-VPS34 (1:1,000, cat#3811), anti-Beclin1 (1:1,000, cat#3738), anti-phospho-p70 S6 Kinase (Thr389) (1:1,000, cat#9205), and anti-mTOR (1:1,000, cat#2983) were obtained from Cell Signaling Technology. Anti-Ref2P (1:200, cat#ab178440), anti-GABARAP (1:5,000, cat#ab109364), anti-ATP6V1B2 (1:1,000, cat#ab73404), anti-GM130 (1:5,000, cat#52649), anti-ATP5A (1:1,000, cat#14748), and anti-Rab7 (1:250, cat#137029) were acquired from Abcam. Anti-ubiquitin (1:1,000, cat#sc8017), anti-Cathepsin-L (1:500, cat#sc32320), anti-Myc (1:1,000, cat#sc40), anti-GFP (1:100, cat#sc-9996), anti-V1H (1:100, cat#sc-166227), anti-LAMP2 (1:500, cat#sc-18822), and anti-p70-S6 Kinase (1:500, cat#sc-8418) were from Santa Cruz. Anti-V0D (1:100, cat#18274-1-AP) and anti-Coro1C (1:1,000, cat#14749-1-AP) were from Proteintech. Anti-insect Cathspsin-L (1:500, cat#MAB22591) was from R&D

System. Anti-LC3 (1:2,500, cat#NB100-2220) was from Novus. Anti-p62 (1:1,000, cat#PM045) was from MBL. Anti-Coro1B (1:500, cat#GTX66270) was obtained from Genetex. Anti-actin (1:500, JLA20) was from Developmental Studies Hybridoma Bank (DSHB). Anti-Tubulin (1:5,000, cat#T6074), anti-Flag-M2 (1:1,000, cat#F1804), and anti-HA (1:1,000, cat#H9658) were from Sigma-Aldrich. Anti-USP45 (1:500, cat#PA070065) was from Cusabio.

## Immunoprecipetion and immunoblotting

For immunoprecipitation, cells were scraped from dishes and lysed in ice-cold lysis buffer (50 mM Tris–HCl, pH 7.4, 150 mM NaCl, 1 mM EDTA, 10% glycerol, 1% Triton X-100, 10 mM NaF, and protease inhibitor cocktail) for 15 min. The cell lysate was then incubated with the indicated antibodies overnight at 4°C, followed by incubation with protein G-Sepharose beads for 1 h. Samples were then washed with cold wash buffer (20 mM HEPES, pH 7.4, 150 mM NaCl, 1.5 mM MgCl, 10% glycerol, and 0.1% Triton X-100), mixed with SDS sample buffer, and then heated at 100°C for 10 min. For the ubiquitination assay under SDS denaturing conditions, cells were lysed with lysis buffer supplemented with 1% SDS. The lysate was then sonicated and incubated at 100°C for 5 min. Subsequently, the sample was diluted to a final concentration of 0.2% SDS for immunoprecipitation, as described above. For LysoIP, cells were transfected with Flag- or HA-tagged TMEM192. After transfection, the cells were washed with PBS and then scraped into 1 ml of KPBS (136 mM KCl, 10 mM $KH_2PO_4$, pH 7.25, adjusted with KOH) and centrifuged at 1,000 $g$ for 2 min at 4°C. The pellet was resuspended in 950 μl KPBS and 25 μl was collected for the whole-cell fraction. The remaining cells were gently homogenized with 20 strokes of a 2 ml homogenizer. The homogenate was then centrifuged at 1,000 $g$ for 2 min at 4°C and the supernatant was incubated with anti-HA magnetic beads (Thermo Fisher Scientific) for 3 min. The precipitated sample was then washed twice with KPBS, mixed with SDS sample buffer, and boiled for 10 min. Lysates were resolved by SDS-polyacrylamide gel electrophoresis (SDS-PAGE) and transferred to PVDF membranes. Membranes were blocked in 5% skim milk in PBS with 0.1% Tween-20 at room temperature for 30 min and incubated with primary antibodies overnight at 4°C. The membranes were then washed several times and incubated with secondary HRP-conjugated antibodies for 1 h at room temperature. Finally, immunoblots were detected with an ECL reagent (Millipore).

## Transmission electron microscopy

Larval fat bodies were dissected and fixed with 2.5% glutaraldehyde and 4% paraformaldehyde in 0.1 M cacodylate buffer (pH 7.4) at 4°C overnight. The samples were washed three times with 0.1 M cacodylate buffer (15 min per wash), followed by post-fixation in cold buffer containing 1% osmium tetroxide and 1.5% potassium hexacyanoferrate(II) for 30 min. After postfixation, samples were incubated in $H_2O$ with 1% thiocarbohydrazide for 30 min at 30°C. The second fixation was performed with 1% osmium tetroxide, followed by incubation with 2% uranyl acetate for 30 min. The samples were then dehydrated in a graded series of ethanol, embedded in Spurr's resin, and sectioned using

a Leica EM UC7 ultramicrotome. The slices were stained with 4% uranyl acetate and 0.5% lead citrate, and images were captured with an FEI Tecnai G2-F20 S-TWIN field emission gun (FEG) transmission electron microscope.

## RT-PCR
Total RNA was extracted from the larval fat body or mammalian cells following the GENEzol TriRNA Pure Kit (Geneaid) protocol. cDNA was synthesized from 2 µg of RNA using the iScript cDNA Synthesis Kit (Bio-Rad). For RT-PCR, 1 µg of cDNA was used as the template and mixed with primers and 2× SuperRed PCR Master Mix (Biotools). The primers used were as below: 5′-CCA CGATTGCTCAACGGATT-3′ (dUsp45-L), 5′-CGTAGAGTAGAC CCGTGGAC-3′ (dUsp45-R), 5′-TGGTGGTGGAACTTTCAAGGC-3′ (USP45-L), 5′-AGGTGCCTTCTGACAAAGCTGA-3′ (USP45-R), 5′-TTGTCTGGGCAAGAGGATCAG-3′ (Actin5C-L), 5′-ACCACTCGC ACTTGCACTTTC-3′ (Actin5C-R).

## Statistical analysis
Statistical analysis was conducted using Student's $t$-test for comparisons between two groups. For experiments with more than two groups, data were analyzed using one-way ANOVA followed by Tukey's or Dunnett's multiple comparisons test. Survival data were analyzed using log-rank tests. All statistical analyses were performed using GraphPad Prism 10 software. All data are presented as mean ± SEM. Differences were considered significant if P values were <0.05 (*), 0.01 (**), 0.001 (***), or 0.0001 (****).

## Online supplemental material
Fig. S1 shows that dUsp45 impairs autophagy and V-ATPase translocation to the autolysosome; Fig. S2 demonstrates that depletion of USP45 causes enrichment of V-ATPase at the lysosome; Fig. S3 illustrates that USP45 interacts with Coro1B and mediates its ubiquitination; Fig. S4 reveals that USP45 depletion accelerates the recovery of acidification and actin patch structure formation following the removal of the actin polymerization inhibitor, LatA; and Fig. S5 shows that Coro1B knockdown promotes lysosomal function and V-ATPase translocation to the lysosome in an N-WASP-dependent manner.

## Data availability
The data supporting the findings of this study are available in the article and its supplementary files, or from the corresponding author upon request.

## Acknowledgments
We thank Drs. Ruey-Hwa Chen, Thomas Neufeld, Won-Jing Wang, Wei Yuan Yang, the Bloomington Stock Center, Vienna *Drosophila* RNAi Center, and Fly Core Taiwan for reagents and fly stocks; the National RNAi Core Facility at Academia Sinica for shRNAs; Academia Sinica Biological Electron Microscopy Core Facility and Chih-Wei Chang (Chang Gung University) for electron microscopy assistance; the Institute of Biological Chemistry (IBC) Bio-Imaging Core Facility for their assistance with confocal analysis. We are grateful to members of the Chen laboratory for helpful discussions during the course of this work.

This work was supported by the Taiwan National Science and Technology Council (MOST111-2311-B-001-018-MY3) and by Academia Sinica. Open Access funding provided by the National University of Taiwan.

Author contributions: Y.J. Lin: Conceptualization, Data curation, Formal analysis, Investigation, Methodology, Project administration, Resources, Validation, Visualization, Writing - original draft, Writing - review & editing, L.-T. Huang: Formal analysis, Investigation, Methodology, Validation, Visualization, P.-Y. Ke: Investigation, G.-C. Chen: Conceptualization, Data curation, Funding acquisition, Methodology, Project administration, Resources, Supervision, Validation, Visualization, Writing - original draft, Writing - review & editing.

Disclosures: The authors declare no competing interests exist.

Submitted: 2 July 2024

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

**Supplemental material**

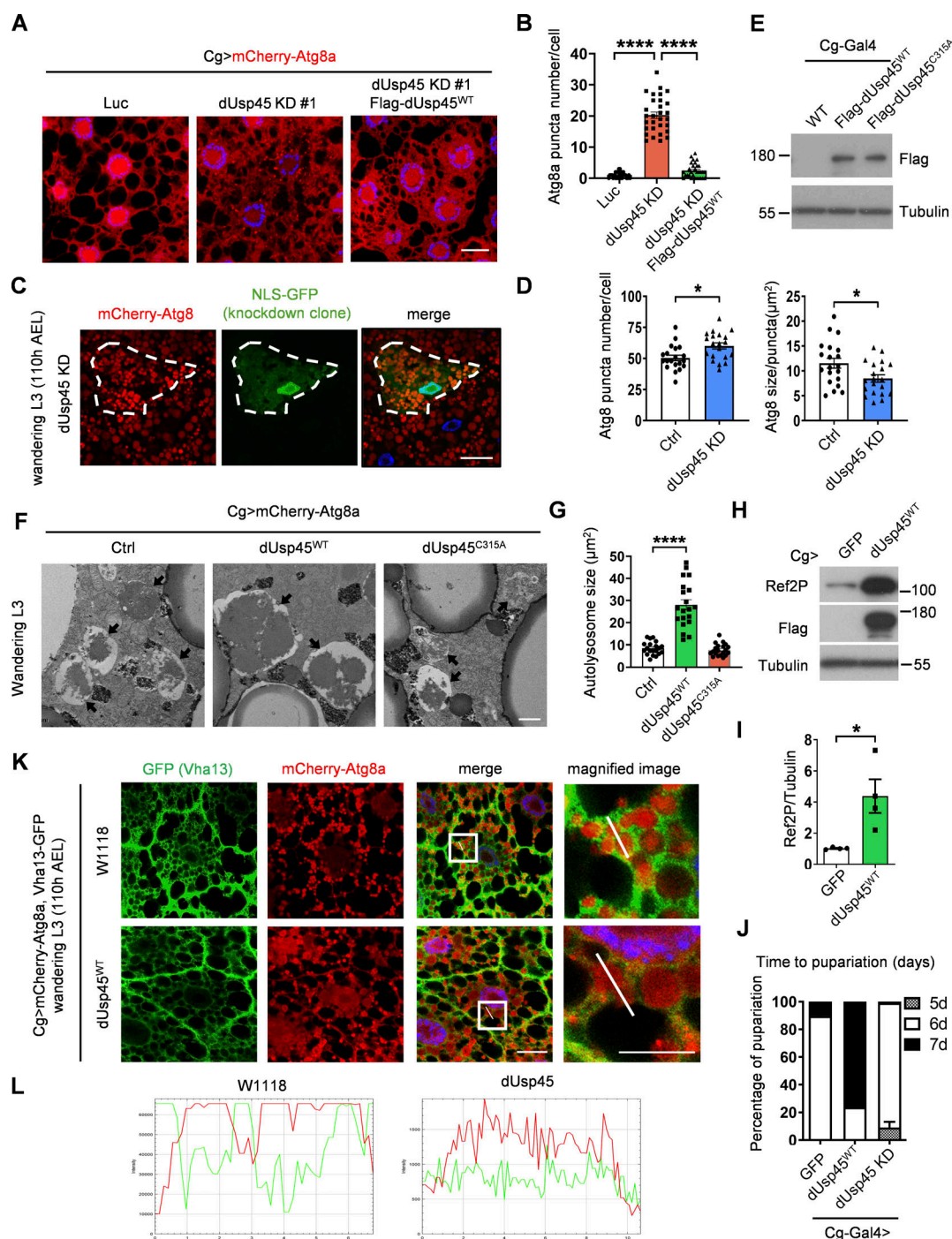

Figure S1. **dUsp45 impairs autophagy and V-ATPase translocation to the autolysosome. (A)** Confocal microscopic analysis of mCherry-Atg8a puncta in control (Luc), *dUsp45* knockdown, and *dUsp45* rescue (dUsp45 depletion with dUsp45^WT over-expression) larvae fat body. Scale bar, 20 μm. **(B)** Quantification of Atg8a puncta number in A. Data are shown as mean ± SEM in *n* = 3, ≥30 cells. **(C)** Clonal analysis of wL3 larval fat body cells showing mCherry-Atg8 signals in control and *dUsp45* knockdown cells (marked with NLS-GFP). Scale bar, 20 μm. **(D)** Quantification of Atg8 puncta size in control and *dUsp45* knockdown cell shown in C. Data shown as mean ± SEM, *n* = 3, ≥ 20 cells. **(E)** Western blot analysis of expression level of wild-type (WT) and catalytic mutant (C315A) of dUsp45 in larvae fat body indicated by Flag antibody. **(F)** Representative TEM images showed the ultrastructure of control (Luciferase), dUsp45 wild-type (WT), and catalytic mutant (CA) overexpression wL3 larval fat body cells. Black arrow indicates autolysosome. Scale bar, 1 μm. **(G)** Quantification of autolysosome size in F; data shown as mean ± SEM, *n* = 3, ≥ 20 autolysosomes. **(H)** Western blot analysis of Ref2P expression in wandering larvae fat body with control and dUsp45^WT overexpression. **(I)** Quantification of Ref2P was normalized to Tubulin. Data are shown as mean ± SEM in four of independent experiments. **(J)** Diagram shows the percentage of control, dUsp45 overexpression, and knockdown larvae pupariated at indicated days. The data were collected from ≥90 flies in each group. **(K)** Confocal microscopy analysis of Vha13 (GFP-positive) and mCherry-Atg8a localization in control and dUsp45^WT-overexpressing wL3 larval fat body cells. Scale bar, 20 μm (original) and 10 μm (zoom-in). **(L)** Line-scan profiles of fluorescence intensity for mCherry-Atg8a and GFP-Vha13 along the white line in K. Significance was determined using one-way ANOVA and Tukey's (B) or Dunnett's (G) multiple comparisons test, and Student's *t* test (D and I); *P < 0.05; ****P < 0.0001. Source data are available for this figure: SourceData FS1.

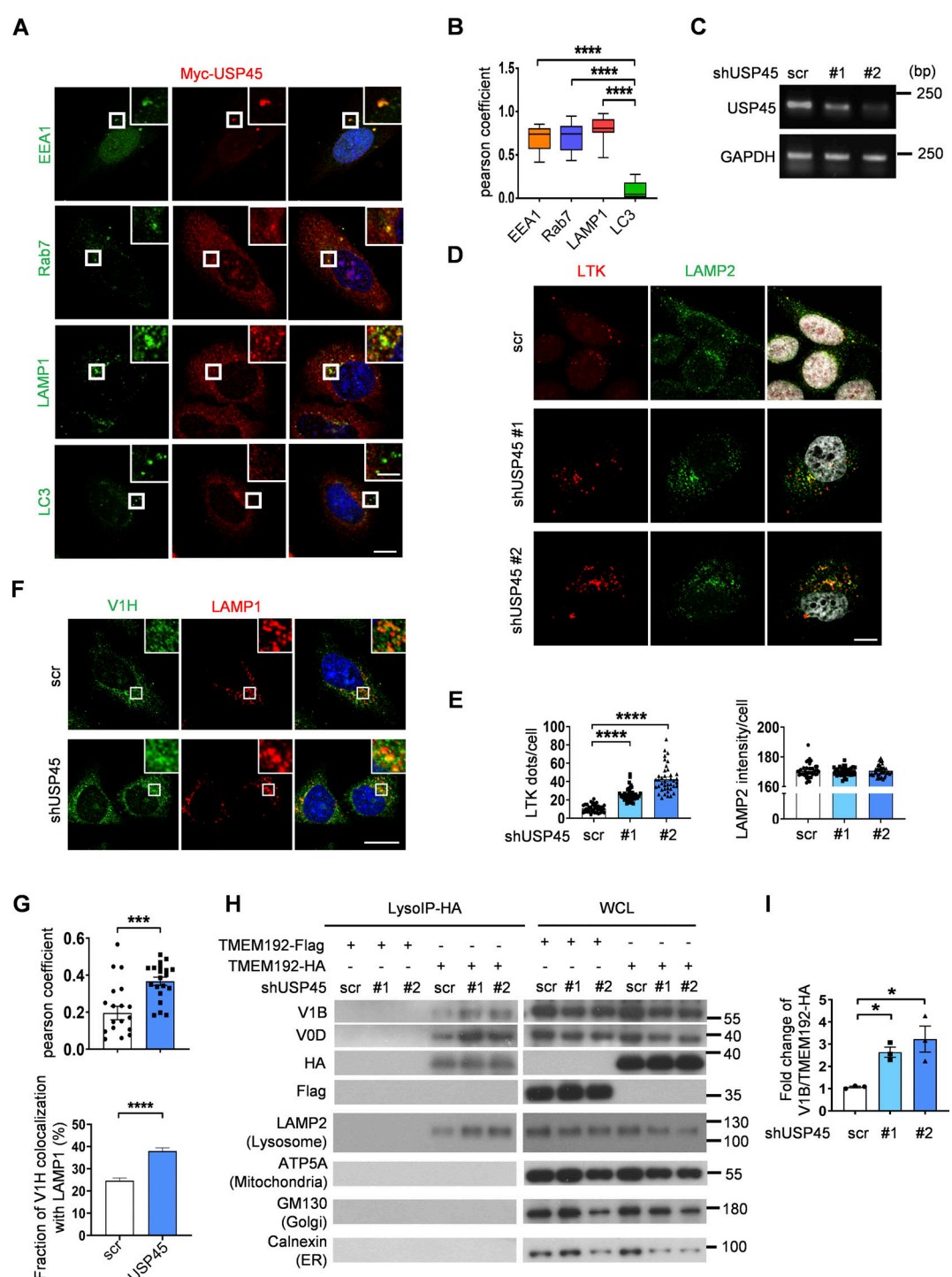

Figure S2. **Depletion of USP45 causes enrichment of V-ATPase at the lysosome. (A)** Immunofluorescent analysis of the colocalization of transiently transfected Myc-USP45 and endogenous vesicle markers (EEA1, Rab7, LAMP1, and LC3) in HeLa cells using the indicated antibodies. Scale bar, 10 µm (original) and 4 µm (zoom-in). **(B)** Quantification of colocalization of Myc-USP45 and markers in A. Data are shown as mean ± SEM in *n* = 3, ≥15 cells from three independent experiments. **(C)** RT-PCR analysis of *USP45* expression in control and *USP45* knockdown HeLa cells. **(D)** Immunofluorescent analysis of Lyso-Tracker (LTK) and lysosome (LAMP2) in *USP45* knockdown cells. Scale bar, 10 µm. **(E)** Quantification of LTK dot number and LAMP2 intensity respectively in D. Data shown as mean ± SEM in *n* = 3, ≥40 cells. **(F)** Immunofluorescent analysis of colocalization of V-ATPase subunit (V1H) and lysosome (LAMP1). Scale bar, 20 µm. **(G)** Quantification of colocalization of V1H and LAMP1 in F. ± SEM in *n* = 3, ≥20 cells. **(H)** Lysosomal immunoprecipitation (LysoIP) analysis of V-ATPase subunits (V1Band V0D) in enriched lysosome lysate by precipitating TMEM192-HA (TMEM192-Flag as negative control) in control and *USP45* knockdown HEK293T cells. V-ATPase subunits (V1B, V0D) and organelle markers (lysosome, LAMP2; mitochondria, ATP5A; Golgi, GM130; and ER, Calnexin) were detected by antibodies. **(I)** Quantification of V1B normalized to HA. Data are shown as mean ± SEM from three independent experiments. Significance was determined using one-way ANOVA and Tukey's (B) or Dunnett's (E and I) multiple comparisons test, and Student's *t* test (G); *P < 0.05; ***P < 0.001; ****P < 0.0001. Data shown as mean Source data are available for this figure: SourceData FS2.

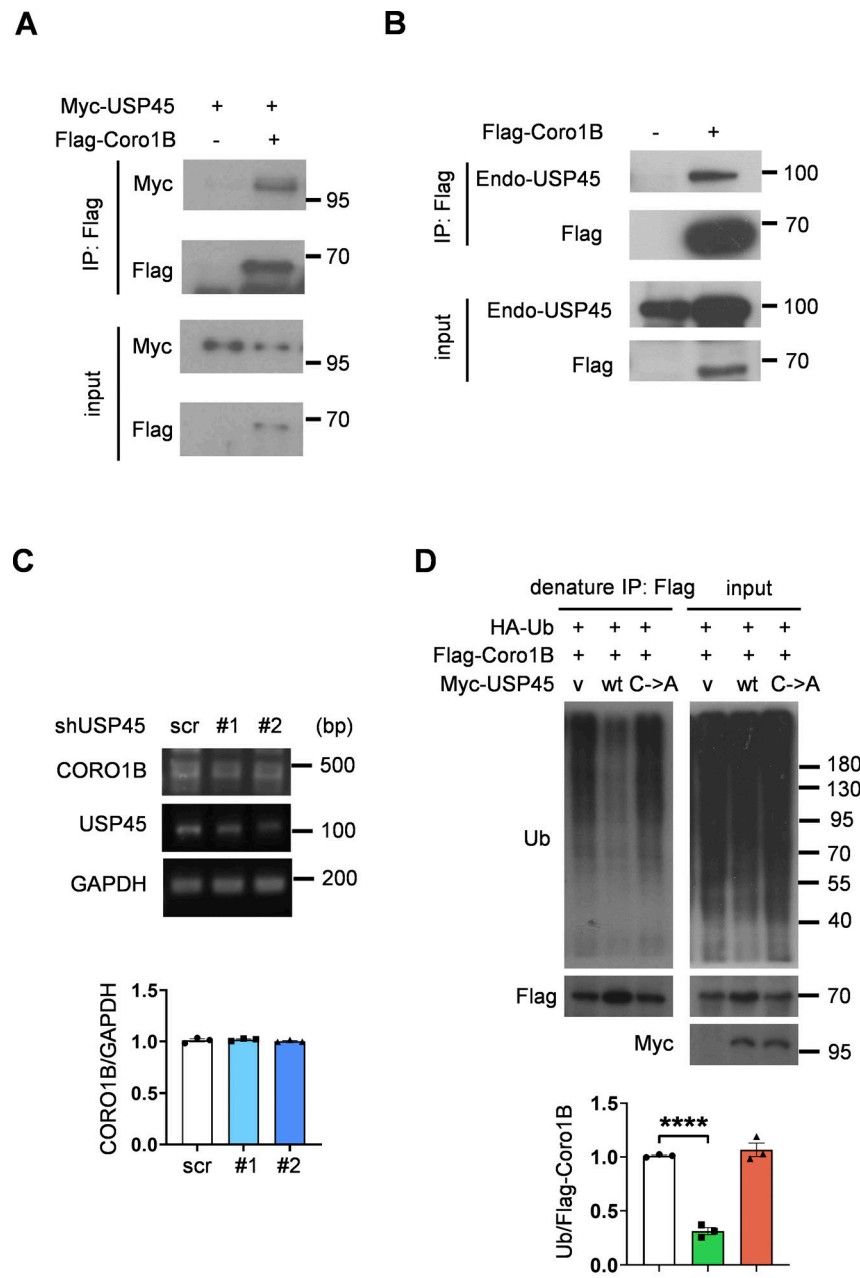

Figure S3.   **USP45 interacts with Coro1B and mediates its ubiquitination. (A)** Immunoprecipitation analysis of interaction between Flag-Coro1B and Myc-USP45 with Flag antibody. **(B)** Immunoprecipitation analysis of the interaction between Flag-Coro1B and endogenous USP45 by pulldown of Flag-Coro1B. **(C)** RT-PCR analysis of *CORO1B* mRNA level in control and *USP45* knockdown HeLa cell. Quantification of *CORO1B* mRNA level was normalized to *GAPDH*. Data are shown as mean ± SEM from three independent experiments. **(D)** Immunoprecipitation analysis for Flag-Coro1B ubiquitination in HEK293T cells with expression of wild-type (WT) or mutant (C199A) Myc-USP45. Cells were lysed by a denaturing agent (1% SDS) containing buffer. Quantification of Ub was normalized to Flag-Coro1B. Data are shown as mean ± SEM from three independent experiments. Significance was determined using one-way ANOVA and Dunnett's multiple comparisons test (C and D); ****P < 0.0001. Source data are available for this figure: SourceData FS3.

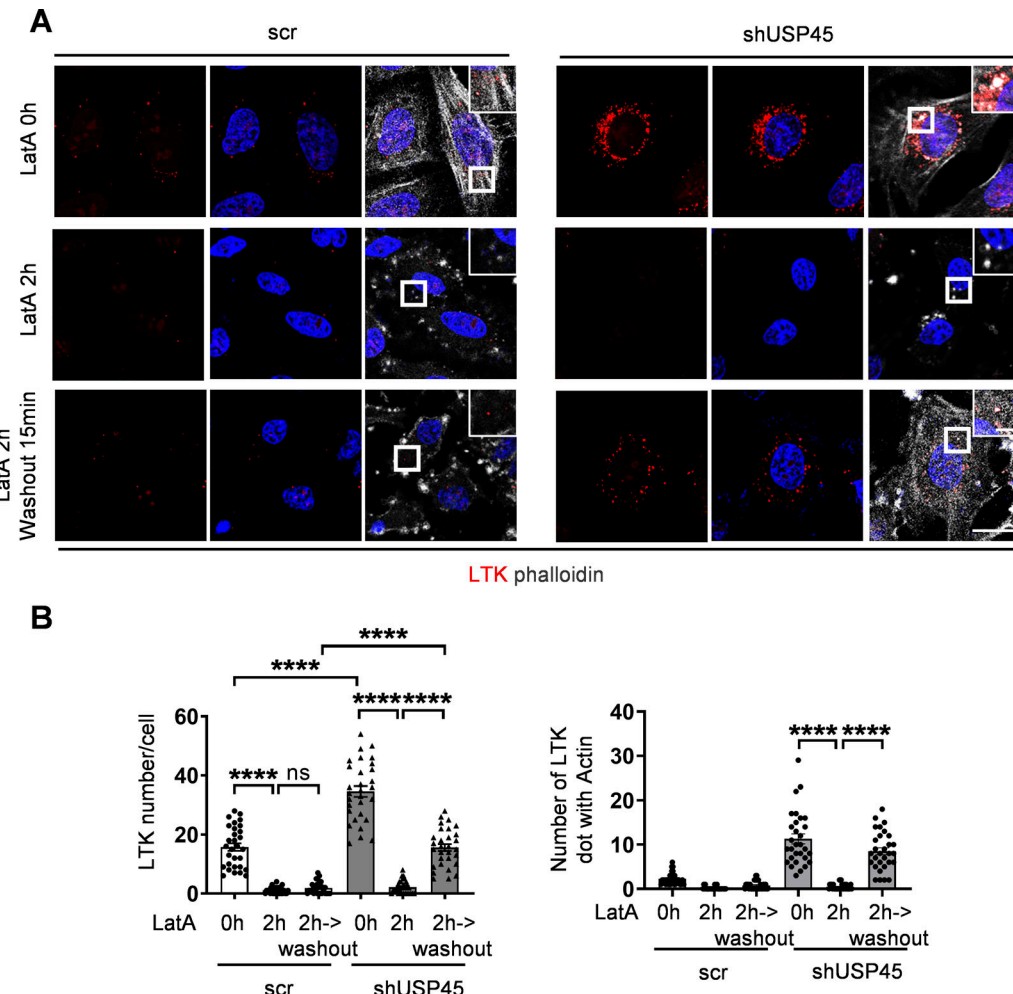

Figure S4. **USP45 depletion accelerates recovery of acidification and actin patch structure formation after removal of actin polymerization inhibitor, LatA. (A)** Immunofluorescent analysis of LysoTracker (LTK) and actin structure (phalloidin) in control and USP45 knockdown HeLa cells. Cells were treated with LatA (200 nM, 2 h) followed by replacement of fresh medium for 15 min. Scale bar, 20 µm (original) and 5 µm (zoom-in). **(B)** Quantification of LTK dot number and LTK-phalloidin colocalization in A. Data are shown as mean ± SEM in $n$ = 3, ≥30 cells. Significance was determined using one-way ANOVA and Tukey's multiple comparisons test; ****P < 0.0001.

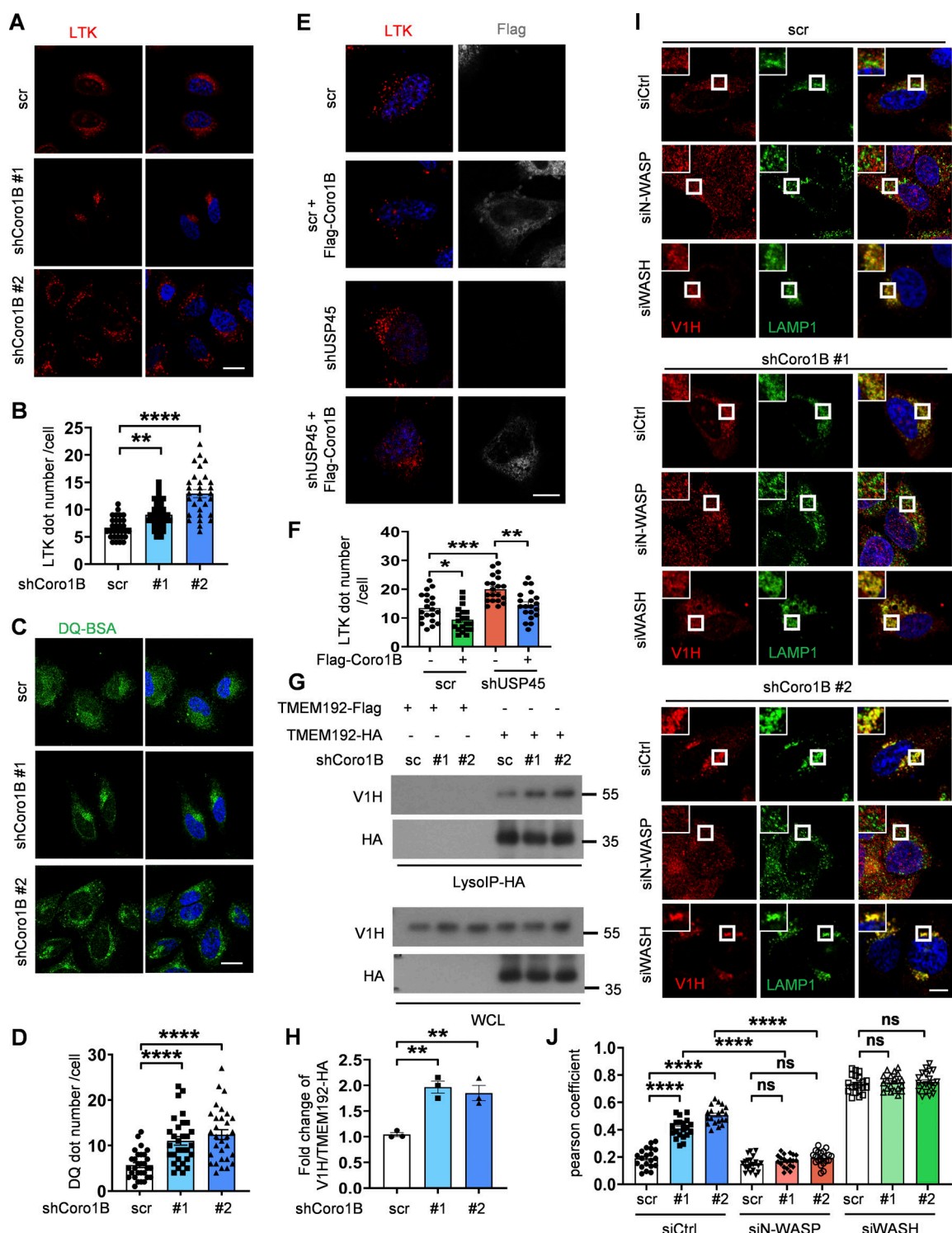

Figure S5. **CORO1B knockdown promotes lysosomal function and V-ATPase translocation to lysosome in an N-WASP-dependent manner. (A)** Immunofluorescent analysis of LysoTracker (LTK) in control and *CORO1B* knockdown HeLa cell. Scale bar, 10 μm. **(B)** Quantification of LTK dot number in A. Data are shown as mean ± SEM in *n* = 3, ≥30 cells. **(C)** Confocal microscopic analysis of DQ-BSA in control and *CORO1B* knockdown cells. Scale bar, 10 μm. **(D)** Quantification of DQ-BSA number in C. Data are shown as mean ± SEM in *n* = 3 , ≥30 cells. **(E)** Confocal microscopic analysis of LTK signals in control and *USP45* knockdown cells with or without Flag-Coro1B overexpression. Scale bar, 10 μm. **(F)** Quantification of LTK dot number in E. Data shown as mean ± SEM in *n* = 3, ≥20 cells. **(G)** Lysosomal immunoprecipitation (LysoIP) analysis of V-ATPase subunit (V1H) in enriched lysosome lysate of control and *CORO1B* knockdown HEK293T cells. TMEM192-Flag was emerged as negative control. **(H)** Quantification of V1H normalized to HA in G. Data shown as mean ± SEM in three of independent experiments. **(I)** Immunofluorescent analysis of the colocalization of the V-ATPase subunit (V1H) and lysosome marker (LAMP1) in control and Coro1B-depleted HeLa cells with knockdown of control or WASP family proteins (N-WASP, WASH) by the indicated siRNAs. **(J)** Quantification of colocalization of V1H and LAMP1 in I. Data are shown as mean ± SEM in *n* = 3, ≥20 cells. Significance was determined using one-way ANOVA and Dunnett's (B, D, and H) or Tukey's (F and J) multiple comparisons test; **P < 0.01; ***P < 0.001; ****P < 0.0001. Source data are available for this figure: SourceData FS5.

