## [Peer Review File · The Journal of Cell Biology]

The deubiquitinase USP45 inhibits autophagy through actin regulation by Coronin 1B

Yuchieh Lin, Li-Ting Huang, Po-Yuan Ke, and Guang-Chao Chen

Corresponding Author(s): Guang-Chao Chen, Institute of Biological Chemistry, Academia Sinica

Review Timeline:

Submission Date:	2024-07-02
Editorial Decision:	2024-08-06
Revision Received:	2024-12-20
Editorial Decision:	2025-01-21
Revision Received:	2025-01-26

Monitoring Editor: Hong Zhang

Scientific Editor: Tim Fessenden

Transaction Report:

DOI: <https://doi.org/10.1083/jcb.202407014>

August 6, 2024

Re: JCB manuscript #202407014

Dr. Guang-Chao Chen
Institute of Biological Chemistry, Academia Sinica
128 Sec2 Academia Rd
Taipei 115
Taiwan

Dear Dr. Chen,

Thank you for submitting your manuscript entitled "The deubiquitinase USP45 regulates autophagy and lysosomal activity by modulating actin dynamics". The manuscript was assessed by expert reviewers, whose comments are appended to this letter. We invite you to submit a revision if you can address the reviewers' key concerns, as outlined here.

As you will see, reviewers overall appreciated the novel insights on USP45 and regulation of lysosome acidification. We concur with Reviewers 1 and 3 who suggest adjusting the conclusions to focus the role of this deubiquitinase in V-ATPase targeting to lysosomes, whereas subsequent effects on autophagy are more expected. Reviewer 2 sought clarity into the specific mechanism by which regulation of actin dynamics by Coronin, which may have widespread effects throughout the cell, specifically affects V-ATPase trafficking. We agree this point is central to the main advance in this study and it must be addressed in a revised manuscript with additional data, rather than by toning down conclusions. A suitably revised manuscript must also address points 2 and 4 by Reviewer 1, and points 3 and 4 from Reviewer 2.

GENERAL GUIDELINES:

Text limits: Character count for an Article is < 40,000, not including spaces. Count includes title page, abstract, introduction, results, discussion, and acknowledgments. Count does not include materials and methods, figure legends, references, tables, or supplemental legends.

Figures: Articles may have up to 10 main text figures. Figures must be prepared according to the policies outlined in our Instructions to Authors, under Data Presentation, <https://jcb.rupress.org/site/misc/ifora.xhtml>. All figures in accepted manuscripts will be screened prior to publication.

*****IMPORTANT:** It is JCB policy that if requested, original data images must be made available. Failure to provide original images upon request will result in unavoidable delays in publication. Please ensure that you have access to all original microscopy and blot data images before submitting your revision. ***

Supplemental information: There are strict limits on the allowable amount of supplemental data. Articles may have up to 5 supplemental figures. Up to 10 supplemental videos or flash animations are allowed. A summary of all supplemental material should appear at the end of the Materials and methods section.

Please note that JCB now requires authors to submit Source Data used to generate figures containing gels and Western blots with all revised manuscripts. This Source Data consists of fully uncropped and unprocessed images for each gel/blot displayed in the main and supplemental figures. Since your paper includes cropped gel and/or blot images, please be sure to provide one Source Data file for each figure that contains gels and/or blots along with your revised manuscript files. File names for Source Data figures should be alphanumeric without any spaces or special characters (i.e., SourceDataF#, where F# refers to the associated main figure number or SourceDataFS# for those associated with Supplementary figures). The lanes of the gels/blots should be labeled as they are in the associated figure, the place where cropping was applied should be marked (with a box), and molecular weight/size standards should be labeled wherever possible.

The typical timeframe for revisions is three to four months. While most universities and institutes have reopened labs and

allowed researchers to begin working at nearly pre-pandemic levels, we at JCB realize that the lingering effects of the COVID-19 pandemic may still be impacting some aspects of your work, including the acquisition of equipment and reagents. Therefore, if you anticipate any difficulties in meeting this aforementioned revision time limit, please contact us and we can work with you to find an appropriate time frame for resubmission. Please note that papers are generally considered through only one revision cycle, so any revised manuscript will likely be either accepted or rejected.

Thank you for this interesting contribution to Journal of Cell Biology. You can contact us at the journal office with any questions at cellbio@rockefeller.edu.

Sincerely,

Hong Zhang
Monitoring Editor
Journal of Cell Biology

Tim Fessenden
Scientific Editor
Journal of Cell Biology

Reviewer #1 (Comments to the Authors (Required)):

This study reveals that the deubiquitinating enzyme dUsp45/USP45 is a crucial regulator of autophagy and lysosomal function in *Drosophila* and mammalian cells. Depletion of dUsp45/USP45 stabilizes the actin-binding protein Coronin 1B (Coro1B), which promotes F-actin patch formation and V-ATPase translocation to lysosomes, resulting in increased lysosomal acidification and enhanced autophagy. Additionally, loss of dUsp45/USP45 extends lifespan and reduces polyglutamine-induced toxicity. Overall, this is an interesting paper, and for the most part, the data are compelling. I have a few suggestions to further improve this manuscript:

1. The authors convincingly show that dUsp45/USP45 regulates lysosomal acidification, but they should clarify how this protein also activates autophagy and enhances autophagic flux. The authors should either provide the molecular mechanism for autophagy activation or deemphasize this aspect and focus more on lysosomal acidification.
2. In mammalian cells, the authors need to examine the effect of USP45 knockdown on p62 levels to provide definitive evidence for enhanced autophagic degradation.
3. It should be determined whether USP45 plays a role in starvation-induced degradation.
4. The data quality in Figure 3D needs to be improved.

Reviewer #2 (Comments to the Authors (Required)):

Chen and colleagues discover that the deubiquitinase Usp45 influences autophagy, and conclude that this occurs because of apparently impacting lysosomal activity, actin dynamics, and in turn, organismal health. The authors use robust fly genetics and mammalian cell biology approaches, and their conclusions are largely supported by their data. The authors are encouraged to consider a few issues raised below, and to adjust conclusions in an appropriate manner to be consistent with their data.

- 1) The key mechanistic conclusion is that Usp45 regulates Coro to influence transport of V-ATPase and lysosomal function. Perhaps this reviewer is reading too much into this conclusion, but this seems overstated. This is not the only possible mechanism for V-ATPase "loading" into lysosomes, and although the data support this possible mechanism, this could also be explained by a very indirect impact of the actin cytoskeleton on V-ATPase and lysosome function. Therefore, I encourage the authors not to overstate the breadth of this observation, but rather consider alternative possibilities, including that the cytoskeleton can have very pleiotropic roles in influencing this cell biology. The alternative would be to invest more effort into deriving data suggesting that this is a universal and key regulatory mechanism that support the conclusion that is made.
- 2) The data in support of this being a developmentally controlled process is very minimal. The simple experiment of quantifying Usp45 RNA at 2 widely different developmental times, when physiology is vastly different for multiple documented reasons, and

then making this conclusion could be misleading. The authors should consider that Usp45 could be regulated at multiple levels, could be pleiotropic, and be cautious to assume that the difference that is observed only supports their model.

3) The reliance of overexpression experiments for some conclusions should be approached with appropriate caution. For example, the decision to focus TEM experiments in Figure 1 on overexpression of Usp45 is surprising. If the authors want to use TEM to explore the function of Usp45 at high resolution, they should start by comparison of control and mutant/knockdown. Over-expression can be extremely misleading because of multiple reasons, including that protein overexpression alone can activate cell stress and autophagy, and when you consider over-expression of an enzyme that influences ubiquitin the possible explanations for phenotypes can be numerous. If such data is to be included, many controls and caveats must be considered and stated.

4) The protein physical interaction data is not based on endogenous proteins, and therefore, appropriate caution should be considered in interpretation of these results.

Reviewer #3 (Comments to the Authors (Required)):

In the present study, Chen and coworkers utilized the *Drosophila* system to identify the deubiquitinating enzyme (DUB), dUsp45/USP45, as a critical player in regulating autophagy and lysosomal activity. This has also been validated in mammalian cells. Evidence indicates that loss of dUsp45/USP45 results in autophagy activation measured by the accumulation of fluorescently labeled Atg8-structures. Several assays, including TEM, have validated this, indicating the accumulation of autolysosome vesicles. The authors also provide evidence for the involvement of USP45 in the regulation of translocation of the V-ATPase subunit to the lysosomes, whereby overexpression leads to enhanced lysosomal acidification. Knockdown of USP45 leads to deacidification of the lysosome. Following previously reported studies, they found that Coronin 1B (Coro1B), known to regulate actin depolymerization, needed for translocation of the V-ATPase to the lysosome, is a substrate of USP45. Accordingly, loss of dUsp45/USP45 leads to the formation of F-actin patches and the translocation of V-ATPase to lysosomes in an N-WASP-dependent manner. Finally, the authors provide evidence for the positive effects of dUsp45 depletion on extending lifespan and ameliorating polyglutamine (polyQ)-induced toxicity in *Drosophila*. Overall, the data presented in this study are rather convincing, and the experiments are well-controlled. The novelty stems from the discovery of the activity of the USP45 in regulating lysosomal acidification. My main concern, however, is the author's definition of USP45 as a negative regulator of autophagy. Inhibition of lysosomal acidification will affect the lysosomal activity needed for the degradation of autophagic bodies and other cargo delivered by other pathways, such as endocytosis. The authors do not provide evidence for a direct link to autophagy; therefore, USP45 should be referred to as a regulator of lysosomal activity.

MS ID#: JCB manuscript #202407014

MS TITLE: The deubiquitinase USP45 regulates lysosomal activity by modulating actin dynamics

Reviewer #1 (Comments to the Authors (Required)):

This study reveals that the deubiquitinating enzyme dUsp45/USP45 is a crucial regulator of autophagy and lysosomal function in *Drosophila* and mammalian cells. Depletion of dUsp45/USP45 stabilizes the actin-binding protein Coronin 1B (Coro1B), which promotes F-actin patch formation and V-ATPase translocation to lysosomes, resulting in increased lysosomal acidification and enhanced autophagy. Additionally, loss of dUsp45/USP45 extends lifespan and reduces polyglutamine-induced toxicity. Overall, this is an interesting paper, and for the most part, the data are compelling. I have a few suggestions to further improve this manuscript:

1. The authors convincingly show that dUsp45/USP45 regulates lysosomal acidification, but they should clarify how this protein also activates autophagy and enhances autophagic flux. The authors should either provide the molecular mechanism for autophagy activation or deemphasize this aspect and focus more on lysosomal acidification.

Response: We thank the reviewer for the positive and constructive comments. We agree that the precise role of USP45 in regulating autophagy requires further clarification. Our results in *Drosophila* and mammalian cells showed that ablation of dUsp45/USP45 leads to increased autophagosome formation and enhanced autophagic flux. These findings suggest a direct involvement of USP45 in autophagy, as evidenced by the increased accumulation of autophagic vesicles. To explore the mechanism by which USP45 regulates autophagy, we examined the effects of USP45 depletion on key autophagy-related proteins. Notably, while the expression of mTORC1 was unaffected, we observed a significant reduction in the phosphorylation of its downstream target, p70 S6 kinase (S6K), indicating reduced mTOR activity (see new Fig. 3H-I). This reduction in mTOR activity is known to be associated with autophagy activation. In addition, USP45 depletion resulted in increased expression of ULK1, VPS34, and Beclin1 (see new Fig. 3J-K), which are key regulators of autophagy initiation and autophagosome formation. We also observed a significant increase in GFP-TFEB nuclear localization in USP45-depleted cells (see new Fig. 3L-M), suggesting that USP45 may regulate autophagy through mTOR-dependent modulation of TFEB, a transcription factor that governs the expression of autophagy and lysosomal genes (Napolitano and Ballabio, 2016). These findings support our

hypothesis that USP45 plays a critical role in modulating autophagy by regulating mTOR activity, which in turn influences key autophagy and lysosomal processes.

2. In mammalian cells, the authors need to examine the effect of USP45 knockdown on p62 levels to provide definitive evidence for enhanced autophagic degradation.

Response: We thank the reviewer for the suggestion. We performed Western blot analysis and observed decreased p62 levels in USP45-depleted cells (see new Fig. 3D).

3. It should be determined whether USP45 plays a role in starvation-induced degradation.

Response: We thank the reviewer for the suggestion. Our results showed that the number of LC3 puncta was similar in USP45-depleted cells under both fully fed and nutrient-deprived conditions, suggesting that USP45 knockdown did not further enhance starvation-induced autophagy.

4. The data quality in Figure 3D needs to be improved.

Response: We have provided new Figure 3D.

Reviewer #2 (Comments to the Authors (Required)):

Chen and colleagues discover that the deubiquitinase Usp45 influences autophagy, and conclude that this occurs because of apparently impacting lysosomal activity,

actin dynamics, and in turn, organismal health. The authors use robust fly genetics and mammalian cell biology approaches, and their conclusions are largely supported by their data. The authors are encouraged to consider a few issues raised below, and to adjust conclusions in an appropriate manner to be consistent with their data.

1) The key mechanistic conclusion is that Usp45 regulates Coro to influence transport of V-ATPase and lysosomal function. Perhaps this reviewer is reading too much into this conclusion, but this seems overstated. This is not the only possible mechanism for V-ATPase "loading" into lysosomes, and although the data support this possible mechanism, this could also be explained by a very indirect impact of the actin cytoskeleton on V-ATPase and lysosome function. Therefore, I encourage the authors not to overstate the breadth of this observation, but rather consider alternative possibilities, including that the cytoskeleton can have very pleiotropic roles in influencing this cell biology. The alternative would be to invest more effort into deriving data suggesting that this is a universal and key regulatory mechanism that support the conclusion that is made.

Response: We thank the reviewer for this thoughtful comment. We agree that while our data suggest a role for Usp45 in regulating Coro1B and influencing V-ATPase trafficking and lysosomal function, it is important to be cautious not to overstate our conclusions. As the reviewer notes, the actin cytoskeleton has pleiotropic effects on a wide range of cellular processes, and our proposed mechanism—Usp45 regulating Coro1B-mediated actin dynamics to influence V-ATPase transport—represents one plausible model, supported by our data. However, we acknowledge that this may not be the only mechanism involved. In light of this, we have revised the manuscript to emphasize that further studies are needed to fully understand the role of the actin cytoskeleton in V-ATPase trafficking and lysosomal function, and to consider alternative mechanisms that could also contribute to these processes.

2) The data in support of this being a developmentally controlled process is very minimal. The simple experiment of quantifying Usp45 RNA at 2 widely different developmental times, when physiology is vastly different for multiple documented reasons, and then making this conclusion could be misleading. The authors should consider that Usp45 could be regulated at multiple levels, could be pleiotropic, and be cautious to assume that the difference that is observed only supports their model.

Response: We thank the reviewer for raising this important point. We acknowledge that the data currently presented in support of the developmental regulation of Usp45 are limited, and we appreciate the concern that further analysis is needed to substantiate this claim. We agree that Usp45 expression could be regulated at multiple levels during development and that Usp45 may have pleiotropic effects depending on the developmental stage. The transcriptional profile of Usp45 across different stages of *Drosophila* development shows high expression during embryogenesis, moderate to low expression during the larval stages, and a gradual increase during aging in adults (Graveley et al., 2011). In response to the reviewer's concern, we have revised the manuscript to more cautiously present our conclusions, focusing specifically on Usp45's role in regulating developmental autophagy in *Drosophila* wandering third-instar (wL3) larval fat body tissues. Consistently, we observed a reduction in *dUsp45* gene expression in the fat bodies of wL3 larvae compared to second-instar (L2) larvae (Fig. 1F). We further examined whether *dUsp45* plays a role in the regulation of developmental autophagy. As observed in L2 larvae, *dUsp45* depletion caused smaller autolysosomes in the fat body cells of wL3 larvae compared to control cells (new Fig. S1C-D). We recognize that further investigation is necessary to fully understand the mechanisms regulating Usp45 expression and its pleiotropic effects across development. Additionally, our new data from multiple time points show that *dUsp45* mRNA expression is significantly upregulated during aging in adult flies (new Fig. 8), which coincides with a decline in autophagy/lysosomal function.

3) The reliance of overexpression experiments for some conclusions should be approached with appropriate caution. For example, the decision to focus TEM experiments in Figure 1 on overexpression of Usp45 is surprising. If the authors want to use TEM to explore the function of Usp45 at high resolution, they should start by comparison of control and mutant/knockdown. Over-expression can be extremely misleading because of multiple reasons, including that protein overexpression alone can activate cell stress and autophagy, and when you consider over-expression of an enzyme that influences ubiquitin the possible explanations for phenotypes can be numerous. If such data is to be included, many controls and caveats must be considered and stated.

Response: We thank the reviewer for this insightful comment. We fully agree that overexpression experiments should be interpreted with caution, particularly when investigating the function of Usp45. As the reviewer points out, overexpression of a protein can induce cellular stress and activate compensatory pathways, which may

confound the interpretation of the results. In our initial TEM experiments, we focused on overexpression of Usp45 to explore the potential ultrastructural changes associated with elevated levels of the protein. In response to the reviewer's suggestion, we have now included an additional control condition using a catalytically inactive Usp45-CA mutant in the TEM experiments. These updated data are now presented in Supplementary Figure S1F-G. We believe this additional control helps to address the potential confounding effects of Usp45 overexpression and provides further context for interpreting the observed phenotypes. Furthermore, we have revised the discussion to emphasize that while overexpression can provide valuable insights into the potential roles of Usp45, the most robust conclusions about its physiological function should be drawn from loss-of-function and mutant analyses.

4) The protein physical interaction data is not based on endogenous proteins, and therefore, appropriate caution should be considered in interpretation of these results.

Response: We thank the reviewer for this important comment. Unfortunately, commercially available Coro1B and USP45 antibodies do not yield reliable results for immunoprecipitation of endogenous proteins. To address this concern, we performed additional experiments where we used Flag-tagged Coro1B to immunoprecipitate endogenous USP45. These results, shown in new Figure S4B, support the interaction between endogenous USP45 and Flag-Coro1B. We hope this provides additional confidence in the validity of our findings, despite the limitations of commercially available antibodies for direct immunoprecipitation of the endogenous proteins.

Reviewer #3 (Comments to the Authors (Required)):

In the present study, Chen and coworkers utilized the *Drosophila* system to identify the deubiquitinating enzyme (DUB), dUsp45/USP45, as a critical player in regulating autophagy and lysosomal activity. This has also been validated in mammalian cells. Evidence indicates that loss of dUsp45/USP45 results in autophagy activation measured by the accumulation of fluorescently labeled Atg8-structures. Several assays, including TEM, have validated this, indicating the accumulation of autolysosome vesicles. The authors also provide evidence for the involvement of USP45 in the regulation of translocation of the V-ATPase subunit to the lysosomes, whereby overexpression leads to enhanced lysosomal acidification. Knockdown of

USP45 leads to deacidification of the lysosome. Following previously reported studies, they found that Coronin 1B (Coro1B), known to regulate actin depolymerization, needed for translocation of the V-ATPase to the lysosome, is a substrate of USP45. Accordingly, loss of dUsp45/USP45 leads to the formation of F-actin patches and the translocation of V-ATPase to lysosomes in an N-WASP-dependent manner. Finally, the authors provide evidence for the positive effects of dUsp45 depletion on extending lifespan and ameliorating polyglutamine (polyQ)-induced toxicity in *Drosophila*.

Overall, the data presented in this study are rather convincing, and the experiments are well-controlled. The novelty stems from the discovery of the activity of the USP45 in regulating lysosomal acidification. My main concern, however, is the author's definition of USP45 as a negative regulator of autophagy. Inhibition of lysosomal acidification will affect the lysosomal activity needed for the degradation of autophagic bodies and other cargo delivered by other pathways, such as endocytosis. The authors do not provide evidence for a direct link to autophagy; therefore, USP45 should be referred to as a regulator of lysosomal activity.

Response: We thank the reviewer for the positive and constructive comments. We agree that the precise role of USP45 in regulating autophagy requires further clarification. Our data in both *Drosophila* and mammalian cells demonstrate that depletion of dUsp45/USP45 leads to increased autophagosome formation and enhanced autophagic flux. These findings suggest a direct involvement of USP45 in autophagy, as evidenced by the increased accumulation of autophagic vesicles. To explore the mechanism by which USP45 regulates autophagy, we examined the effects of USP45 depletion on key autophagy-related proteins. Notably, while the expression of mTORC1 was unaffected, we observed a significant reduction in the phosphorylation of its downstream target, p70 S6 kinase (S6K), indicating reduced mTOR activity (see new Fig. 3H-I). This reduction in mTOR activity is known to be associated with autophagy activation. In addition, USP45 depletion resulted in increased expression of ULK1, VPS34, and Beclin1 (see new Fig. 3J-K), which are key regulators of autophagy initiation and autophagosome formation. We also observed a significant increase in GFP-TFEB nuclear localization in USP45-depleted cells (see new Fig. 3L-M), suggesting that USP45 may regulate autophagy through mTOR-dependent modulation of TFEB, a transcription factor that governs the expression of autophagy and lysosomal genes (Napolitano and Ballabio, 2016). These findings support our hypothesis that USP45 plays a critical role in modulating

autophagy by regulating mTOR activity, which in turn influences key autophagy and lysosomal processes.

References

- Graveley, B.R., A.N. Brooks, J.W. Carlson, M.O. Duff, J.M. Landolin, L. Yang, C.G. Artieri, M.J. van Baren, N. Boley, B.W. Booth, J.B. Brown, L. Cherbas, C.A. Davis, A. Dobin, R. Li, W. Lin, J.H. Malone, N.R. Mattiuzzo, D. Miller, D. Sturgill, B.B. Tuch, C. Zaleski, D. Zhang, M. Blanchette, S. Dudoit, B. Eads, R.E. Green, A. Hammonds, L. Jiang, P. Kapranov, L. Langton, N. Perrimon, J.E. Sandler, K.H. Wan, A. Willingham, Y. Zhang, Y. Zou, J. Andrews, P.J. Bickel, S.E. Brenner, M.R. Brent, P. Cherbas, T.R. Gingeras, R.A. Hoskins, T.C. Kaufman, B. Oliver, and S.E. Celniker. 2011. The developmental transcriptome of *Drosophila melanogaster*. *Nature*. 471:473-479.
- Napolitano, G., and A. Ballabio. 2016. TFEB at a glance. *J Cell Sci*. 129:2475-2481.

January 21, 2025

RE: JCB Manuscript #202407014R

Guang-Chao Chen
Institute of Biological Chemistry, Academia Sinica

Dear Dr. Chen:

Thank you for submitting your revised manuscript entitled "The deubiquitinase USP45 regulates lysosomal activity by modulating actin dynamics". As you will see, all reviewers recommend publication with no further changes. We would be happy to publish your paper in JCB pending final revisions necessary to meet our formatting guidelines (see details below).

A. MANUSCRIPT ORGANIZATION AND FORMATTING:

Full guidelines are available on our Instructions for Authors page, <http://jcb.rupress.org/submission-guidelines#revised>. Submission of a paper that does not conform to JCB guidelines will delay the acceptance of your manuscript.

1) Text limits: Character count for Articles is < 40,000, not including spaces. Count includes abstract, introduction, results, discussion, and acknowledgments. Count does not include title page, figure legends, materials and methods, references, tables, or supplemental legends.

2) Figures limits: Articles may have up to 10 main figures and 5 supplemental figures/tables.

3) Figure formatting: Scale bars must be present on all microscopy images, including inset magnifications. Molecular weight or nucleic acid size markers must be included on all gel electrophoresis. Please avoid pairing red and green for images and graphs to ensure legibility for color-blind readers. If red and green are paired for images, please ensure that the particular red and green hues used in micrographs are distinctive with any of the colorblind types. If not, please modify colors accordingly or provide separate images of the individual channels.

** Please include size markers in Figure 1F, 8C, and 8E.

4) Statistical analysis: Error bars on graphic representations of numerical data must be clearly described in the figure legend. The number of independent data points (n) represented in a graph must be indicated in the legend. Statistical methods should be explained in full in the materials and methods. For figures presenting pooled data the statistical measure should be defined in the figure legends. Please also be sure to indicate the statistical tests used in each of your experiments (either in the figure legend itself or in a separate methods section) as well as the parameters of the test (for example, if you ran a t-test, please indicate if it was one- or two-sided, etc.). Also, if you used parametric tests, please indicate if the data distribution was tested for normality (and if so, how). If not, you must state something to the effect that "Data distribution was assumed to be normal but this was not formally tested."

** Please indicate the statistical tests used in all figure legends.

5) Abstract and title: The abstract should be no longer than 160 words and should communicate the significance of the paper for a general audience. The title should be less than 100 characters including spaces. Make the title concise but accessible to a general readership.

** The revised manuscript extends findings from lysosomal acidification to regulation of autophagy more broadly. Meanwhile, the regulation of Coronin 1B is clearly established however this work does not focus on actin polymerization/depolymerization itself.

In light of this we suggest the title should be modified to:

"The deubiquitinase USP45 promotes autophagy through actin regulation by Coronin1B"

We also request changes to the abstract accordingly, which should refer to regulation of actin structures by Coronin 1B rather than actin dynamics.

6) Materials and methods: Should be comprehensive and not simply reference a previous publication for details on how an experiment was performed. Please provide full descriptions in the text for readers who may not have access to referenced manuscripts. We also provide a report from SciScore and an associate score, which we encourage you to use as a means of evaluating and improving the methods section.

7) Please be sure to provide the sequences for all of your primers/oligos, plasmids, and RNAi constructs in the materials and methods. You must also indicate in the methods the source, species, and catalog numbers (where appropriate) for all of your antibodies. Please also indicate the acquisition and quantification methods for immunoblotting/western blots.

8) Microscope image acquisition: The following information must be provided about the acquisition and processing of images:

- Make and model of microscope
- Type, magnification, and numerical aperture of the objective lenses
- Temperature
- Imaging medium
- Fluorochromes
- Camera make and model
- Acquisition software
- Any software used for image processing subsequent to data acquisition. Please include details and types of operations involved (e.g., type of deconvolution, 3D reconstitutions, surface or volume rendering, gamma adjustments, etc.).

10) Supplemental materials: There are strict limits on the allowable amount of supplemental data. Articles may have up to 5 supplemental figures. Please also note that tables, like figures, should be provided as individual, editable files. A summary of all supplemental material should appear at the end of the Materials and methods section.

13) ORCID IDs: ORCID IDs are unique identifiers allowing researchers to create a record of their various scholarly contributions in a single place. At resubmission of your final files, please provide an ORCID ID for all authors.

15) A data availability statement is required for all research article submissions. The statement should address all data underlying the research presented in the manuscript. Please visit the JCB instructions for authors for guidelines and examples of statements at (<https://rupress.org/jcb/pages/editorial-policies#data-availability-statement>).

Please note that JCB requires authors to submit Source Data used to generate figures containing gels and Western blots with all revised manuscripts. This Source Data consists of fully uncropped and unprocessed images for each gel/blot displayed in the main and supplemental figures. Since your paper includes cropped gel and/or blot images, please be sure to provide one Source Data file for each figure that contains gels and/or blots along with your revised manuscript files. File names for Source Data figures should be alphanumeric without any spaces or special characters (i.e., SourceDataF#, where F# refers to the associated main figure number or SourceDataFS# for those associated with Supplementary figures). The lanes of the gels/blots should be labeled as they are in the associated figure, the place where cropping was applied should be marked (with a box), and molecular weight/size standards should be labeled wherever possible. Source Data files will be directly linked to specific figures in the published article.

Journal of Cell Biology now requires a data availability statement for all research article submissions. These statements will be published in the article directly above the Acknowledgments. The statement should address all data underlying the research presented in the manuscript. Please visit the JCB instructions for authors for guidelines and examples of statements at (<https://rupress.org/jcb/pages/editorial-policies#data-availability-statement>).

B. FINAL FILES:

Thank you for your attention to these final processing requirements. Please revise and format the manuscript and upload materials within 7 days. If you need an extension for whatever reason, please let us know and we can work with you to determine a suitable revision period.

Thank you for this interesting contribution, we look forward to publishing your paper in Journal of Cell Biology.

Sincerely,

Hong Zhang
Monitoring Editor
Journal of Cell Biology

Tim Fessenden
Scientific Editor
Journal of Cell Biology

Reviewer #1 (Comments to the Authors (Required)):

The authors have answered all of my queries satisfactorily. Congratulations!

Reviewer #2 (Comments to the Authors (Required)):

Chen and colleagues have responded to my comments, and I am satisfied with the revised manuscript.

Reviewer #3 (Comments to the Authors (Required)):

None